



Atmospheric
Chemistry
and Physics

# Dominant role of emission reduction in PM$_{2.5}$ air quality improvement in Beijing during 2013–2017: a model-based decomposition analysis

**Jing Cheng[1], Jingping Su[2], Tong Cui[2], Xiang Li[3], Xin Dong[2], Feng Sun[2], Yanyan Yang[2], Dan Tong[1], Yixuan Zheng[1], Yanshun Li[1], Jinxiang Li[2], Qiang Zhang[1], and Kebin He[1,4]**

[1]Ministry of Education Key Laboratory for Earth System Modeling, Department of Earth System Science, Tsinghua University, Beijing, 100084, China
[2]Beijing Municipal Environmental Monitoring Center, Beijing, 100048, China
[3]Beijing Municipal Bureau of Ecology and Environment, Beijing, 100048, China
[4]State Key Joint Laboratory of Environment Simulation and Pollution Control, School of Environment, Tsinghua University, Beijing 100084, China

**Correspondence:** Qiang Zhang (qiangzhang@tsinghua.edu.cn) and Jinxiang Li (jinxiangli@hotmail.com)

**Abstract.** In 2013, China's government published the Air Pollution Prevention and Control Action Plan (APPCAP) with a specific target for Beijing, which aims to reduce annual mean PM$_{2.5}$ concentrations in Beijing to 60 µg m$^{-3}$ in 2017. During 2013–2017, the air quality in Beijing was significantly improved following the implementation of various emission control measures locally and regionally, with the annual mean PM$_{2.5}$ concentration decreasing from 89.5 µg m$^{-3}$ in 2013 to 58 µg m$^{-3}$ in 2017. As meteorological conditions were more favourable to the reduction of air pollution in 2017 than in 2013 and 2016, the real effectiveness of emission control measures on the improvement of air quality in Beijing has frequently been questioned.

In this work, by combining a detailed bottom-up emission inventory over Beijing, the MEIC regional emission inventory and the WRF-CMAQ (Weather Research and Forecasting Model and Community Multiscale Air Quality) model, we attribute the improvement in Beijing's PM$_{2.5}$ air quality in 2017 (compared to 2013 and 2016) to the following factors: changes in meteorological conditions, reduction of emissions from surrounding regions, and seven specific categories of local emission control measures in Beijing. We collect and summarize data related to 32 detailed control measures implemented during 2013–2017, quantify the emission reductions associated with each measure using the bottom-up local emission inventory in 2013, aggregate the measures into seven categories, and conduct a series of CMAQ simulations to quantify the contribution of different factors to the PM$_{2.5}$ changes.

We found that, although changes in meteorological conditions partly explain the improved PM$_{2.5}$ air quality in Beijing in 2017 compared to 2013 (3.8 µg m$^{-3}$, 12.1 % of total), the rapid decrease in PM$_{2.5}$ concentrations in Beijing during 2013–2017 was dominated by local (20.6 µg m$^{-3}$, 65.4 %) and regional (7.1 µg m$^{-3}$, 22.5 %) emission reductions. The seven categories of emission control measures, i.e. *coal-fired boiler control*, *clean fuels in the residential sector*, *optimize industrial structure*, *fugitive dust control*, *vehicle emission control*, *improved end-of-pipe control*, and *integrated treatment of VOCs*, reduced the PM$_{2.5}$ concentrations in Beijing by 5.9, 5.3, 3.2, 2.3, 1.9, 1.8, and 0.2 µg m$^{-3}$, respectively, during 2013–2017. We also found that changes in meteorological conditions could explain roughly 30 % of total reduction in PM$_{2.5}$ concentration during 2016–2017 with more prominent contribution in winter months (November and December). If the meteorological conditions in 2017 had remained the same as those in 2016, the annual mean PM$_{2.5}$ concentrations would have increased from 58 to 63 µg m$^{-3}$, exceeding the target established in the APPCAP. Despite the remarkable impacts from meteorological condition changes, local and regional emission reductions still played major roles in the PM$_{2.5}$ decrease in Beijing during 2016–2017, and

Please note the remarks at the end of the manuscript.

*clean fuels in the residential sector*, *coal-fired boiler control*, and *optimize industrial structure* were the three most effective local measures (contributing reductions of 2.1, 1.9, and 1.5 $\mu g\,m^{-3}$, respectively). Our study confirms the effectiveness of clean air actions in Beijing and its surrounding regions and reveals that a new generation of control measures and strengthened regional joint emission control measures should be implemented for continued air quality improvement in Beijing because the major emitting sources have changed since the implementation of the clean air actions.

## 1   Introduction

Most countries inevitably undergo and tackle severe air pollution in the development process. In recent years, severe $PM_{2.5}$ pollution in China has gradually become an urgent challenge to the government (Wang and Hao, 2012; Li et al., 2017). It not only posed a threat to human health but has also badly influenced the social economy and ecological environment (Menon et al., 2002; Chan et al., 2006; Ming et al., 2009; Zheng et al., 2015; Zhu et al., 2015). Beijing, as the capital of China, has suffered especially severe air quality problems. In 2013, the annual average $PM_{2.5}$ concentrations in Beijing reached $90\,\mu g\,m^{-3}$, which was nearly 3 times higher than China's National Ambient Air Quality Standard (NAAQS) of $35\,\mu g\,m^{-3}$ (MEP, 2012). In addition to the high annual average $PM_{2.5}$ concentrations, several frequent and severe heavy haze episodes in January 2013 made the situation even worse and caused great public concern (Zhang et al., 2014; G. Zheng et al., 2016).

To address the increasingly serious $PM_{2.5}$ pollution, the Chinese government released the Air Pollution Prevention and Control Action Plan (APPCAP) in September 2013, which aimed to mitigate severe PM pollution across China, especially in some typical regions. In particular, the average $PM_{2.5}$ concentrations of Beijing should be reduced to less than $60\,\mu g\,m^{-3}$ by 2017. Based on the ambition and guidance of the APPCAP, Beijing has made further efforts and formulated the Beijing 2013–2017 Clean Air Action Plan (referred to as the Beijing Action Plan) to mitigate air pollution. The Beijing Action Plan represents the most important and systematic set of local air pollution control and management policies in the past 5 years. After implementing a series of air pollution control policies and measures, the annual mean $PM_{2.5}$ concentrations in Beijing decreased to $58\,\mu g\,m^{-3}$ in 2017 (BMEP, 2018), 35.2 % lower than that in 2013 and surpassing the air quality goals of the APPCAP. Meanwhile, the surrounding regions of Beijing, such as Tianjin, Hebei, Shandong, Shanxi, and Henan province, also implemented the APPCAP, and the air quality of the whole region has attained marked improvements, which have also been confirmed by satellite-based and ground-based observations (Liu

et al., 2016; S. Cai et al., 2017; Zhao et al., 2017; J. Wang et al., 2017; Y. Zheng et al., 2017).

The $PM_{2.5}$ concentrations in the atmosphere are affected by several factors, while pollutant emissions, regional transport, and meteorological conditions play dominant roles (He et al., 2001; Chen et al., 2018; Zhang et al,. 2018a). In general, local pollutant emissions contribute most to the air pollution for a given city, and the control of emissions is always one of the most effective ways to mitigate air pollution. Regarding the influence of regional transport, considering that Beijing is embraced on three sides by mountains except for the south and southeast direction, the transport of air pollutants from the south and southeast can easily affect the $PM_{2.5}$ concentrations in Beijing (Sun et al., 2015; Wang et al., 2015; Chen et al., 2017). However, regional cities in these two directions, such as Baoding, Changzhou, Hengshui, Shijiazhuang, Tangshan, and Tianjin, suffer even worse air pollution (Li et al., 2017). A combination of $PM_{2.5}$ formation and transport resulted in the regionally complex air pollution characteristics in Beijing (Lang et al., 2013; Chen et al., 2016; Y. Zhang et al., 2018; Zhong et al., 2018). Besides the impact of pollutant emissions and transport, $PM_{2.5}$ concentrations are also highly influenced by some other factors, including atmospheric advection, atmospheric diffusion, and secondary aerosol formulation. (Sun et al., 2015; Yin et al., 2016; Zhang et al., 2016). Several studies have also reported that frequent stable meteorological conditions play an important role in severe pollution episodes (Elser et al., 2016; Ma et al., 2017; Zhang et al., 2018b). Based on emission inventories and air quality models, existing studies have established a mature sensitivity decomposition framework to assess the contributions of emission control to air quality improvements (Zhao et al., 2013; W. Cai et al., 2017).

The mitigation of $PM_{2.5}$ pollution in Beijing was significant during 2013–2017, especially during 2016–2017. This impact resulted from the integrated effects of various factors, including the local emission control through the Beijing Action Plan, the surrounding emission reductions through the APPCAP, and the impacts of meteorological condition changes. Several studies have researched the roles of meteorology as well as regional and local emissions in Beijing's $PM_{2.5}$ pollution; however, most of these studies analysed a single factor or focused on heavy pollution episodes (Wang et al., 2013; Y. Wang et al., 2017; Zeng et al., 2014; Zhang et al., 2015; Zheng et al., 2016; Liu et al., 2017; Ma et al., 2017). There was no systematic and decomposed attribution analysis of Beijing's air quality improvements at an annual scale, especially during the periods of 2013–2017 and 2016–2017, in which the $PM_{2.5}$ concentrations in Beijing decreased significantly. To better understand the great progress in the cleaning of air in Beijing in recent years, a more comprehensive analysis covering the periods of 2013–2017 and 2016–2017 is urgently needed. In this study, based on several sensitivity simulations, we established a decomposition analysis framework to evaluate the impacts of local control poli-

cies, surrounding emission reductions, and the meteorological changes on PM$_{2.5}$ abatements in Beijing during 2013–2017 and 2016–2017. First, the emission reductions of Beijing and its surroundings were estimated based on the quan-

5 tification of air pollution control measures; meanwhile, a new multiple-pollutant emission inventory of Beijing and its surroundings, covering the periods of 2013–2017, was updated and developed. Second, based on a zero-out method, we designed a set of sensitivity experiments under different local

and regional emission control measures and different meteorological conditions. Third, we used the Weather Research and Forecasting Model and Community Multiscale Air Quality model to reproduce and simulate the air quality under different meteorological conditions and emission scenarios.

Finally, based on a zero-out approach, an integrated and decomposed attribution analysis of PM$_{2.5}$ abatements in Beijing was developed to quantify the impacts of local pollution control, surrounding emission reductions, and meteorological changes. The study also identified the key point of next

steps for air pollution control, which would be beneficial for future policymaking.

## 2 Methodology and data

A model-based decomposition attribution analysis of PM$_{2.5}$ abatements in Beijing during 2013–2017 and 2016–2017 was

25 developed under the framework shown in Fig. 1. First, we used the observation data from 12 national observation stations in Beijing to review the air quality during 2013–2017, especially the monthly PM$_{2.5}$ concentrations in this period. The contributions of total PM$_{2.5}$ abatements in Beijing in

2017 were decomposed into three basic parts, including meteorology change, surrounding emission control, and local emission control. Then, we used the WRF-CMAQ (Weather Research and Forecasting Model and Community Multiscale Air Quality) modelling system and observed PM$_{2.5}$ con-

centrations to quantify the contributions of these three factors. To further evaluate the effect of local control policies, we divided the Beijing Action Plan into seven specific policy types, estimated the corresponding emission reductions, and updated the emission inventory during 2013–2017 based

on Beijing's local emission inventory (BJ-EI, described in Sect. 2.2) and the framework of the MEIC model (Zhang et al., 2007; http://www.meicmodel.org/, last access: 30 August 2018). The contributions from local emission control to the PM$_{2.5}$ air quality improvements in Beijing were also decom-

posed into specific measures with the WRF-CMAQ model and measure-related sensitivity experiments.

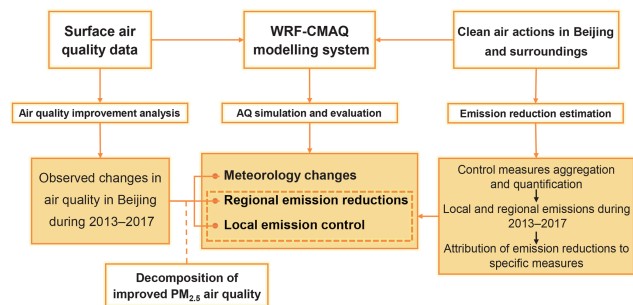

**Figure 1.** Methodological framework for the decomposition analysis of improved PM$_{2.5}$-related air quality in Beijing during 2013–2017.

## 2.1 Ground-based and satellite-based observational data

The study first reviewed the air quality in Beijing from 2013 to 2017 with both ground observation data and satellite ob-

50 servation data. Since 2012, Beijing has maintained an automated air quality monitoring network with 35 stations spatially distributed in the 16 administrative districts and counties (the specific locations of the air quality monitors are shown in the Supplement, Fig. S1). Since 2013, hourly con-

55 centrations of SO$_2$, NO$_2$, CO, PM$_{2.5}$, PM$_{10}$, and O$_3$ have been continuously measured and recorded by Beijing's Environment Protection Bureau (EPB). In this study, we reviewed the changes in SO$_2$, NO$_2$, CO, O$_3$, PM$_{2.5}$, and PM$_{10}$ in Beijing annually during 2013–2017 and analysed the monthly

PM$_{2.5}$ concentrations during this period. The hourly observation data we were CE1 from the 12 national observation stations in Beijing, which are included among the above 35 stations (Fig. S1).

Satellite observation data can provide the objective evidence of the emission reductions and the air quality improvements. Based on satellite observations, we analysed the changes of tropospheric vertical column NO$_2$, boundary layer SO$_2$, and aerosol optical depth (AOD) in Beijing

during 2013–2017 to back up the estimated emission trends and reductions (Sect. 3.5). Based on the Ozone Monitoring Instrument (OMI) sensor, the NO$_2$ tropospheric vertical column density product, DOMINO, is produced by the Royal Netherlands Meteorological Institute (KNMI); and the

SO$_2$ boundary layer vertical column density product, OMI SO$_2$, is produced by NASA. The daily AOD observation data comes from Terra and Aqua, based on the MODIS sensor. To reduce uncertainties in the satellite-based evaluation, DOMINO NO$_2$ and OMI SO$_2$ columns with cloud

fraction higher than 0.3 and surface reflectance higher than 0.3 were discarded. Based on our previous research and methods (Zheng et al., 2016), we estimated the ground-level PM$_{2.5}$ concentrations in Beijing using the satellite-derived AOD data.

To evaluate the accuracy of the $PM_{2.5}$ composition simulation, we collected the $PM_{2.5}$ composition data from the Surface PARTiculate mAtter Network (SPARTAN, https://www.spartan-network.org/, last access: 4 February 2019). SPARTAN is a global, long-term project that observed and analysed the particulate matter mass, water-soluble ions, black carbon, and metals since 2013 (Snider et al., 2015). Major SPARTAN measurements include the Air Photon three-wavelength integrating nephelometer and the Air Photon SS4i automated air sampler. SPARTAN monitors are located in nearly 20 highly populated regions in the world, such as Beijing, Hanoi, Singapore, Dhaka, Pretoria, Toronto, and Bondville. Detailed measurements and site information can be found in Snider's research (Snider et al., 2015, 2016). In this research, we used the reconstructed $PM_{2.5}$ speciation data in the Beijing site (located in the Department of Earth System Science, Tsinghua University) (https://www.spartan-network.org/beijing-china, last access: 4 February 2019) to validate the model simulation results of $PM_{2.5}$ compositions and to verify the variation of $PM_{2.5}$ compositions in Beijing during 2013–2017. Additionally, we collected the observational $PM_{2.5}$ compositions from the reported research (Shao et al., 2018) to verify the simulated variations of $PM_{2.5}$ compositions in the CMAQ model. In Shao's research, the multi-size PM samples were observed and analysed at an urban site at Beijing Normal University, and the chemical composition changes from January 2013 to the winter of 2016–2017 in Beijing were compared.

## 2.2 Estimates of local emission reduction from specific control measures

In this study, the anthropogenic emission inventory of Beijing was provided by the Beijing Municipal Environmental Monitoring Center (BMEMC). Based on the bottom-up method, the BMEMC developed a high-resolution emission inventory for Beijing (BJ-EI) of 2013 and 2017. BJ-EI basically had the same source classification as the MEIC model (described in Sect. 2.3); however, the investigation and calculation process of BJ-EI were conducted at the county level, and those of MEIC were conducted at the provincial level. The power, heating, industry (such as cement, iron, steel, chemical industry, manufacturing industry), and most solvent use (such as vehicle paint, ink, paint, and coating) sectors were treated as point sources, with a higher accuracy of emission facility locations. In addition, fugitive dust emissions, including bare soil dust, road dust, and construction dust, were added in BJ-EI but were missing in the MEIC model because of the lack of activity rate data. More detailed comparisons of BJ-EI and the MEIC model can be found in Table S1. Therefore, the spatial distribution and emission source allocation of BJ-EI were more accurate than those of the MEIC model, which can significantly improve the air quality modelling, especially when modelling with finer resolutions (B. Zheng et al., 2017). Meanwhile, more detailed

and objective activity rate, technology distribution, and removal efficacy data at the county level were collected from BJ-EI, which can largely reduce the uncertainty in estimating the emission reductions of each local control policy.

Beijing started the air quality protection process in 1998 and has focused most on the control of $SO_2$ and $NO_x$ (Lu et al., 2010; Wang et al., 2012; Zhang et al., 2016). The Chinese government released the APPCAP nationwide in 2013 and committed to reducing $PM_{2.5}$ pollution for the first time ever (Zheng et al., 2018b). The APPCAP aimed to reduce the annual $PM_{2.5}$ concentrations of the Beijing–Tianjin–Hebei region by 25 % compared with 2013, and, particularly, the $PM_{2.5}$ concentrations of Beijing should be controlled to less than $60 \, \mu g \, m^{-3}$. To fulfil the air quality targets, Beijing released its own Beijing Action Plan under the framework and guidance of the APPCAP, which contained much more ambitious and stricter control measures than ever before. We summarized and classified all measures in the Beijing Action Plan into seven types, including *coal-fired boiler control, clean fuels in the residential sector, optimize industrial structure, improved end-of-pipe control, vehicle emission control, integrated treatment of VOCs*, and *fugitive dust control*. All the quantifiable control measures were listed in Table 1.

Following our previous studies (Jiang et al., 2015), air pollution control policies and measures could be quantified by adjusting the emission calculation parameters; then the emission reduction associated with a single policy can be estimated from the emission difference between before and after the implementation of this specific policy.

## 2.3 Regional emission inventory data

The air pollutant emission inventory of Beijing's surrounding regions (including Tianjin, Hebei, Henan, Shandong, Shanxi, and Inner Mongolia) for the period of 2013–2017 was obtained from the MEIC model and Zheng's updating work (Zheng et al., 2018b). MEIC is a bottom-up emission inventory model developed for China by Tsinghua University, covering 31 provinces in China and the year range of 1990 up to now. More than 700 emission sources were developed in the MEIC model. The methodologies and data on which the MEIC are based, as well as the continuous updating process of pollutant emission factors, have been introduced in our previous studies (Q. Zhang et al., 2007; Liu et al., 2016; Lei et al., 2011; Li et al., 2014, 2017a, b; Shen et al., 2015; Hong et al., 2017; Qi et al., 2017; B. Zheng et al., 2017). For the updated emission inventory and emission reductions of the surrounding regions, we referred to our previous work (Zheng et al., 2018b). In their work, Zheng et al collected the latest Chinese energy statistics data from the National Bureau of Statistics, the industrial production and technology penetration data, and the unpublished data from the Ministry of Ecology and Environment. Then they estimated and updated China's anthropogenic emissions from 2010 to 2017 under the framework of the MEIC model. Particularly, they

**Table 1.** Summary of emission control measures implemented in the Beijing Action Plan (2013–2017).

| Policy type | Measure ID | Specific control measures | Sectors |
|---|---|---|---|
| 1. Coal-fired boiler control | 1-1 | By the end of 2017, Beijing had closed the four major coal-fired power plants and constructed four great natural gas thermal-power cogeneration centres instead, which reduced coal use by nearly 8.6 million metric tons in total. | Power and heating |
| | 1-2 | Beijing made great efforts to renovate the coal-fired facilities with capacities of less than 25 MW in urban districts and 7 MW in the whole city. A total capacity of 390 000 MW CE2 of coal-fired boilers were eliminated or replaced by clean fuels such as natural gas and electricity. Coal use was reduced by more than 8.5 million metric tons during this process. | Industry |
| 2. Clean fuels in the residential sector | 2-1 | Through cutting down non-peak household power prices and establishing new gas heating systems, approximately 900 000 households in Beijing were converted from using coal to using gas or electricity, and coal use was reduced by a total of 2.1 million metric tons. | Residential |
| | 2-2 | The burning of biomass, such as wood and crops, was thoroughly forbidden by the end of 2016. | Residential |
| 3. Optimize industrial structure | 3-1 | During 2013 to 2017, Beijing phased out a total of 1992 large high-pollution enterprises in the chemical engineering, furniture manufacturing, printing, and non-metal mineral product industries; furthermore, three-quarters of the cement industry was eliminated. | Industry |
| | 3-2 | In the last 5 years, especially since 2016, Beijing has made great efforts to eliminate the small, clustered, and polluting factories that cannot meet efficiency, environmental, and safety standards, and a total of 11 000 such factories were managed or eliminated. | Industry |
| 4. Improved end-of-pipe control | 4-1 | Since 2013, a total of 468 projects involving cleaner production and technological upgrades for high-pollution industrial sectors were carried out by the government. During this process, Beijing has gradually retrofitted all cement factories to achieve thorough denitrification and degassing and has enhanced the desulfurization retrofitting of the non-metal mineral product and chemical engineering industries. | Industry |
| | 4-2 | Beijing first promoted low-nitrogen-burning (LNB) combustion for all industrial sectors in 2013, and the LNB transformation of nearly 21 000 MW of gas-fired boilers and oil-fired boilers has been completed. | Industry |
| 5. Vehicle emission control | 5-1 | (1) During 2013 to 2017, Beijing retired a total of 2 167 000 old vehicles, and all "yellow-labelled" cars (gasoline and diesel cars that failed to meet Euro I and Euro III standards) were eliminated completely by 2017. (2) In March 2017, Beijing completely implemented the China 6/VI fuel quality standard, which is one of the most tightened emission standards in the world. (3) A total of 51 000 taxis completed the replacement of three-way catalytic converters, and 17 000 heavy-duty diesel vehicles were equipped with wall flow particle traps. (4) In September 2017, all out-of-city diesel vehicles with lower emission standards than China III were forbidden to travel within the sixth ring road. | On-road transportation |
| | 5-2 | (1) By the end of 2016, all off-road vehicles were required to comply with "Limits and measurement methods for exhaust pollutants from diesel engines of non-road mobile machinery (China III, IV)". (2) Since 2017, the use of heavily polluting off-road vehicles has been restricted in the six major urban districts and Tongzhou district. | Off-road transportation |
| 6. Integrated treatment of VOCs | 6-1 | Beijing started to eliminate organic solvent coatings, bituminous waterproof materials, and organic painted furniture manufacturing in 2013. Meanwhile, Beijing promoted the use of high-solids and waterborne paints, which contain much fewer organic chemicals, in machine manufacturing, printing, coating, and automobile repair sectors. | Solvent use |
| | 6-2 | During 2013 to 2017, the Yanshan company, the only petrochemical industry enterprise in Beijing, completed seven extensive VOC CE3 control projects, such as the innovation of sealing and defocusing technology, the detection and repair of leakage points, and the specialized management of refined oil production and storage areas. | Industry |
| 7. Fugitive dust control | 7-1 | Beijing increased the quality and frequency of the road cleaning process. By the end of 2017, a mechanized cleaning process was adopted in an area of 90 580 000 $m^2$, accounting for 88 % of the total urban road area. | Fugitive dust |
| | 7-2 | Beijing shut down a total of 310 concrete mixing plants and updated over 20 000 cinder block transporters. Additionally, more than 1200 construction sites were equipped with a video monitoring system at the exits and entrances. | Fugitive dust |
| | 7-3 | By the end of 2015, Beijing completed an afforestation project in advance and afforested nearly 700 $km^2$ in nearby plain areas. | Fugitive dust |

calibrated the emission calculation parameters (such as technology penetrations and removal efficiencies) in major sectors, such as power, cement, steel, and iron, of each province based on emission control policies. Based on their updated China anthropogenic emission inventory, we collected and analysed the detailed sectoral emission reductions and trends of Beijing's surrounding regions during 2013–2017.

## 2.4 WRF-CMAQ model

### 2.4.1 Model configuration

In this work, WRFv3.8 and CMAQ 5.1 were used to build up the air quality modelling system. The WRF model provided the meteorological conditions, while the CMAQ model simulated the air quality and main pollutant concentrations.

For the simulation area, three nested domains were designed in this study (Fig. S2), with a horizontal resolution of 36 km × 36 km, 12 km × 12 km, and 4 km × 4 km, respectively. The first domain covered the entire China area and some parts of south and east Asia; the second one covered the majority of eastern and northern China; and the third one focused on Beijing and its surrounding regions, including the municipality of Tianjin and the provinces of Hebei, Henan, Shandong, Shanxi, and Inner Mongolia. To reduce the uncertainty of meteorological boundary conditions, the simulation size of the WRF model was three grid cells larger than that of the CMAQ model in each domain. The vertical resolution was designed as 23 sigma levels from the surface to tropopause (about 100 mbar) for the WRF simulation (with 10 layers below 3 km), while it collapsed into 14 chemical transport model layers by the Meteorology-Chemistry Interface Processor (MCIP). The 14 sigma levels for the CMAQ model vertical resolution were 1.000, 0.995, 0.988, 0.980, 0.970, 0.956, 0.938, 0.893, 0.839, 0.777, 0.702, 0.582, 0.400, 0.200, and 0.000.

For the WRF model configuration, we chose the New Goddard scheme (Chou et al., 1998) and the rapid radiative transfer model (RRTM; Mlawer et al., 1997) for shortwave and longwave radiation options, the Kain–Fritsch cloud parameterization (version 2, Kain, 2004), the ACM2 PBL scheme (Pleim, 2007), the Pleim–Xiu land-surface scheme (Xiu and Pleim, 2001), and WSM6 cloud microphysics (Hong and Lim, 2006). Analysis nudging, observational nudging, and soil nudging were adopted, and FDDA data were from the US National Centers for Environmental Prediction (NCEP, http://rda.ucar.edu/datasets/, last access: 16 August 2018), Automated Data Processing surface (ds461.0) and upper (ds351.0) air data. The meteorological initial and boundary conditions were derived from the final analysis data (FNL). We made a continuous meteorology simulation during 2013–2017, with a 10 d spin-up before this period.

For the CMAQ model configuration, we applied the CB05 as a gas-phase chemical mechanism and AERO6 as the particulate matter chemical mechanism. The online computation

of photolytic rates was adopted using the simulated aerosol and ozone concentrations. The chemical initial and boundary conditions of the first domain were interpolated from the output of the GEOS-Chem model (Bey et al., 2001; Geng et al., 2015). We firstly simulated a complete time series of all pollutant's concentrations during 2013–2017 (as base cases) and then modelled the 18 sensitivity experiments (described in Sect. 2.5). A 10 d spin-up period was adopted for each sensitivity simulation to mitigate the initial condition impacts. To make the analysis and evaluation more comprehensive, we simulated all sensitivity scenarios for the whole year in cases where the severe pollution period was missing. Similar configurations for the WRF and CMAQ model were applied in our previous studies and exhibited good agreement with observations (B. Zheng et al., 2015; B. Zheng et al., 2017).

For the emission inputs of the model system, the anthropogenic emission inventory for Beijing was taken from BMEMC, and the inventories for other regions in China were provided by the MEIC model, which has been updated to the 2017 level based on Zheng's work (Zheng et al., 2018b). Emissions for other Asian countries were derived from the MIX emission inventory (Model Inter-comparison Study Asia Phase III, MICS-ASIA III; Li et al., 2017a). The biogenic emissions were taken from the Model of Emission of Gases and Aerosols from Nature (MEGAN v2.1). For the dust emission, bare lands dust was calculated by the in-line windblown dust in the CMAQ model. As Sect. 2.2 described, other dust sources, such as road dust and construction dust, were added in BJ-EI, while they were missed in the MEIC model. The lack of fugitive dust in the emission inventory brought uncertainty of air quality simulation, especially for the $PM_{10}$ simulation in other regions (discussed in Sect. 2.4.2).

### 2.4.2 Model validation

To evaluate the meteorology results simulated by the WRF model, we collected the hourly observed meteorology data from the Computational and Information Systems Laboratory at the National Center for Atmospheric Research in Boulder (NCAR, https://rda.ucar.edu/, last access: 16 August 2018) and calculated the mean bias (MB), mean error (ME), correlation coefficient (Corr), root mean square error (RMSE), normalized mean bias (NMB), and normalized mean error (NME). The evaluation results showed that the simulation basically reproduced the meteorological conditions in 2013 and 2017, and the temperature simulation especially featured a high accuracy. The monthly evaluation results of the simulated temperature, relative humidity, wind speed, and wind direction for Beijing in 2013 to 2017 are shown in Table S2a–f.

To evaluate the pollutant concentrations simulated by the CMAQ model, we collected the hourly observed major pollutant concentration data from the Beijing Municipal Environmental Protection Bureau. The decrement of the $PM_{2.5}$ con-

centration is the most important target in APPCAP, as well as the major pollutant this research focused on. We emphatically analysed the accuracy of $PM_{2.5}$ simulations and listed the monthly descriptive statistics (MB, ME, Corr, RMSE, NMB, and NME) of the hourly observational $PM_{2.5}$ and the CMAQ model simulation for 12 national stations in Beijing during 2013–2017 (Table S3a). The time series of $PM_{2.5}$ from observations and CMAQ simulations in three base cases ($E_{L13S13}M_{13}$, $E_{L16S16}M_{16}$, and $E_{L17S17}M_{17}$, described in Sect. 2.5) for Beijing are shown in Fig. 2. Furthermore, we calculated the annual descriptive statistic characters of the observational and simulated $PM_{2.5}$ with other five major pollutants ($SO_2$, $NO_2$, $PM_{10}$, CO, and $O_3$) in Beijing during 2013–2017, which can be found in Table S3b. The time series and evaluation results indicated that the CMAQ model and simulation results in this work can reproduce the temporal and spatial distribution of air pollutants in Beijing and its surroundings relatively well. As for the simulated $PM_{2.5}$ of 2017, the monthly Corr of $PM_{2.5}$ concentrations varied from 0.53 (in May) to 0.89 (in October), and the annual Corr of $PM_{2.5}$ concentrations varied from 0.65 (in 2016) to 0.81 (in 2014). The NMB and NME of monthly $PM_{2.5}$ simulations were CE4 $\pm 45$ % and $\pm 55$ %, respectively. According to the observation data, the annual average $PM_{2.5}$ concentrations in Beijing decreased by 31.5 $\mu g\,m^{-3}$ from 2013 to 2017, while the simulated $PM_{2.5}$ decreased by 32.8 $\mu g\,m^{-3}$ (Table 2). Compared with 2016, the observed and simulated $PM_{2.5}$ decreased by 14.9 and 16.6 $\mu g\,m^{-3}$, respectively (Table 2). The evaluation results suggested that the modelling system in this work can be used to quantify and analyse the attribution of $PM_{2.5}$ mitigation in Beijing. As for the simulation results of other pollutants in Beijing, the Corr varied from 0.61–0.74 for $SO_2$, 0.59–0.68 for $NO_2$, 0.62–0.78 for CO, 0.64–0.74 for $O_3$, and 0.62–0.74 for $PM_{10}$ (Table S3b), which was acceptable for the research. The $SO_2$ simulation was overestimated in the five years, especially during 2013–2015, which indicated the $SO_2$ emission in BJ-EI might be higher than the reality. The added fugitive dust emission (road and construction dust) in BJ-EI has improved the $PM_{10}$ simulation of Beijing noticeably, with an overestimation range of 4–12 % (Table S3b). However, the $PM_{10}$ simulation of other cities in the third domain was underestimated ($-8$ %)–(34 %), especially in some heavy industry cities such as Tangshan ($-34$ %), Baoding ($-25$ %), and Handan ($-29$ %). This might be attributed to the lack of construction and road dust emissions in these regions, as well as the uncertainty of the dust model (Todd et al., 2008; Foroutan et al., 2017). It might introduce the uncertainty of simulation, but, given that our research was focused on the attribution analyses of anthropogenic emission changes in Beijing, this uncertainty was relatively small. The $O_3$ was underestimated in this WRF-CMAQ model system, with the range of ($-8.3$ %)–($-22.6$ %). The rough vertical layers, the underestimation of nature source emissions, the defect of upper boundary simulation in the regional model, and the uncer-

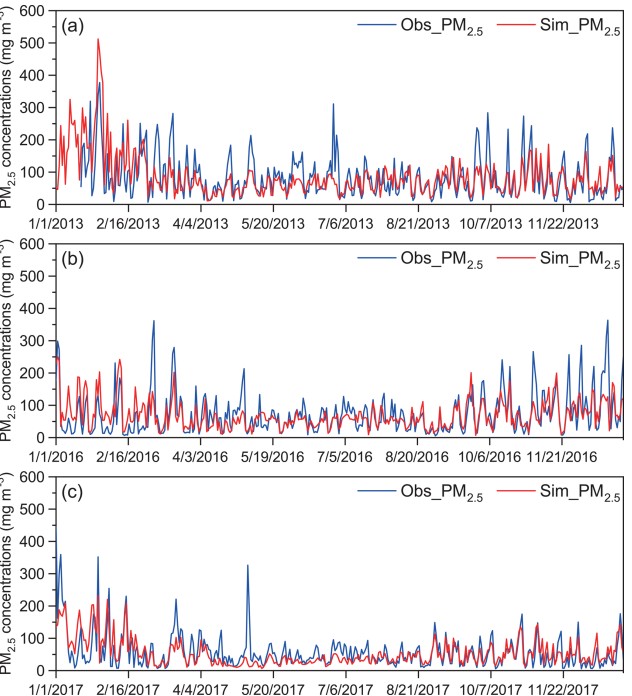

**Figure 2.** Comparison of observed (blue) and CMAQ-simulated (red) daily mean $PM_{2.5}$ concentrations over Beijing in 2013 **(a)**, 2016 **(b)**, and 2017 **(c)**. Observation data were obtained and averaged from 12 national observation stations in Beijing. Simulated concentrations were extracted from the grids corresponding to the station locations.

tainties of VOC (volatile organic compound) emission inventories might all lead this underestimation.

Besides $PM_{2.5}$ concentrations, we also evaluated the simulated $PM_{2.5}$ compositions. We compared the $PM_{2.5}$ compositions of the observations from the Beijing site, SPARTAN, and the simulations from the same grid, which can be found in Table S4. Generally speaking, secondary inorganic aerosol (SIA) was overestimated in most periods (0.1–56.6 %), especially for the nitrate ($NO_3^-$) simulation, which overestimated 6.56–89.9 %. Organic matter (OM) was underestimated in most periods, with the underestimation range of ($-3.0$ %)–($-41.8$ %). This might be caused by the missing mechanism and insufficient simulation of secondary organic aerosol (SOA) formulations in the CMAQ model. The NMB of simulated and observed black carbon (BC) varied from ($-52.9$ %) to (38.2 %). Similar evaluation results can be found in Weagle's research (Weagle et al., 2018), where he compared the GEOS-Chem simulation with SPARTAN observation data, with an average 45.7 % overestimation of SIA, $-58.9$ % underestimation of BC, and $-19.2$ % underestimation of OM. However, the variation trends of simulated $PM_{2.5}$ compositions were basically consistent with the SPARTAN data; both had the remarkable decrement in OM and increment in $NO_3^-$. In Shao's observational results, the $SO_4^{2-}$ proportion in the winter of 2016–2017 in Beijing re-

**Table 2.** CMAQ simulations conducted in this study for the decomposition analysis. The seven types of emission control policies include *coal-fired boiler control* (policy 1), *clean fuels in the residential sector* (policy 2), *optimize industrial structure* (policy 3), *improved end-of-pipe control* (policy 4), *vehicle emission control* (policy 5), *integrated treatment of VOCs* (policy 6), and *fugitive dust control* (policy 7). The parentheses on rightmost column indicate the annual mean observational PM$_{2.5}$ concentrations of 2013 (89.5), 2016 (72.9), and 2017 (58.0).

| Case label | Year of meteorological data in Beijing | Year of emission data in Beijing | Year of emission data in surrounding regions | Purpose of the simulation | Simulated PM$_{2.5}$ (µg m$^{-3}$) |
|---|---|---|---|---|---|
| $E_{L13S13M13}$ | 2013 | 2013 | 2013 | reproduce air quality in 2013 | 86.3 (89.5) |
| $E_{L16S16M16}$ | 2016 | 2016 | 2016 | reproduce air quality in 2016 | 70.1 (72.9) |
| $E_{L17S17M17}$ | 2017 | 2017 | 2017 | reproduce air quality in 2017 | 53.5 (58.0) |
| $E_{L17S17M13}$ | 2013 | 2017 | 2017 | quantify the impact of meteorology compared with 2013 | 57.4 |
| $E_{L17S17M16}$ | 2016 | 2017 | 2017 | quantify the impact of meteorology compared with 2016 | 57.8 |
| $E_{L17S13M17}$ | 2017 | 2017 | 2013 | quantify the contribution from the emission reduction of surroundings during 2013–2017 | 60.9 |
| $E_{L17S16M17}$ | 2017 | 2017 | 2016 | quantify the contribution from the emission reduction of surroundings during 2016–2017 | 55.9 |
| $E_{Lpi S17 M17}(i = 1 - 7)$ | 2017 | 2017 (excluding the implementation of each control policy since 2013) | 2017 | quantify the contribution of each control policy during 2013–2017 | (Supplement) |
| $E_{Lgi S17 M17}(i = 1 - 7)$ | 2017 | 2017 (excluding the implementation of each control policy since 2016) | 2017 | quantify the contribution of each control policy during 2016–2017 | (Supplement) |

duced by 11 % compared with January 2013, while $NO_3^-$ and $NH_4^+$ proportions increased by 77.9 % and 47.3 % (Shao et al., 2018). And in our research, the relative change ratio of $SO_4^{2-}$, $NO_3^-$, and $NH_4^+$ proportions in Beijing from January 2013 to the winter of 2016–2017 were −29.1 %, 89.2 %, and 11.7 %, respectively. In general, the simulated compositions basically captured the variations in observation results, which can support the reasonable analyses of the chemical composition changes.

## 2.5 Scenario design and decomposition analysis

To decompose the attribution of PM$_{2.5}$ abatements in Beijing from 2013 to 2017 and from 2016 to 2017, we set up 18 sensitivity simulations based on a zero-out approach and quantified the contributions of meteorology changes, emission reductions in surrounding areas, and seven types of local emission control policies. Given that the nonlinearity between the response of PM$_{2.5}$ concentrations and the meteorology or emission changes, we discussed uncertainties and limitations in Sect. 3.5. The description and details of all scenarios were listed in Table 2, and the direct simulation results were listed in Table 2, under the column "Simulated PM$_{2.5}$ ($\mu$g m$^{-3}$)".

All scenario cases were labelled as $E_{LiSj}M_k$. $M_k(k)$ represents the meteorological period the case adopted and $E_{LiSj}(i, j)$ represents the emission period. Total emission inventories of China consisted of two parts, the BJ-EI from BMEMC and the regional (all parts of China except for Beijing) emission inventories from the MEIC model. The adopted emission periods of these two parts were labelled as $Li(i)$ and $Sj(j)$, respectively.

$E_{L13S13}M_{13}$, $E_{L16S16}M_{16}$, and $E_{L17S17}M_{17}$ were three base cases, driven by the actual emission inventories and meteorology of 2013, 2016, and 2017, respectively, to reproduce the air quality of the corresponding year. $E_{L17S17}M_{13}$ and $E_{L17S17}M_{16}$ were designed to investigate the impact of meteorology. These two cases were driven by varying meteorological conditions (meteorology of 2013 and 2016, respectively) and the same emission inventory (for the year 2017). $E_{L17S13}M_{17}$ and $E_{L17S16}M_{17}$ were designed to quantify the impact of surrounding emission reduction during 2013–2017 and 2016–2017. In these two cases, the emission inventory of Beijing was set to the 2017 level, while the regional emission inventory was set to the 2013 and 2016 levels, respectively.

Another 14 simulations were designed to quantify the air quality improvements contributed by seven types of local control policies during two periods. Cases for 2013–2017 and 2016–2017 were labelled as $E_{L_{pi}S17}M_{17}$ and $E_{L_{qi}S17}M_{17}$, respectively, where $i$ represents the number of each policy (described and listed in Table 1). The meteorological conditions and regional emission inventories of these 14 cases were set to 2017. For each simulation, emission reductions introduced by the corresponding policy type were added to the 2017 baseline, which is the equivalent of "turn-ing off" this type of policy during this period. And then the derived emission inventory was applied to drive the corresponding air quality modelling.

A linear additive relationship was assumed among all contributors to perform a decomposition analysis, and the simulated contributions of all sensitivity cases were then normalized by the difference in observed PM$_{2.5}$ concentrations from 2013 to 2017 and from 2016 to 2017. The normalization process of the 2013–2017 period was calculated by the following equations, while the simulated results for the period of 2016–2017 can be normalized with the similar process.

$$SCon(M) = SPM_{2.5}(E_{L17S17}M_{13}) − SPM_{2.5}(E_{L17S17}M_{17}), \quad (1)$$

$$SCon(S) = SPM_{2.5}(E_{L17S13}M_{17}) − SPM_{2.5}(E_{L17S17}M_{17}), \quad (2)$$

$$SCon(pi) = SPM_{2.5}(E_{L_{pi}S17}M_{17}) − SPM_{2.5}(E_{L17S17}M_{17}), \quad (3)$$

$$NCon(M) = (PM_{2.5OBS2013} − PM_{2.5OBS2017})$$
$$\times \frac{SCon(M)}{SCon(M) + SCon(S) + \sum_{i=1}^{7}SCon(pi)}, \quad (4)$$

$$NCon(S) = (PM_{2.5OBS2013} − PM_{2.5OBS2017})$$
$$\times \frac{SCon(S)}{SCon(M) + SCon(S) + \sum_{i=1}^{7}SCon(pi)}, \quad (5)$$

$$NCon(pi) = (PM_{2.5OBS2013} − PM_{2.5OBS2017})$$
$$\times \frac{SCon(pi)}{SCon(M) + SCon(S) + \sum_{i=1}^{7}SCon(pi)}, \quad (6)$$

where SCon($M$) represents the simulated contribution of meteorology change during 2013–2017, which equals the balance of simulated PM$_{2.5}$ ($\mu$g m$^{-3}$) from case $E_{L17S17}M_{13}$ and case $E_{L17S17}M_{17}$. Similarly, SCon($M$) and SCon($pi$) represent the simulated contributions of regional emission reductions and each local control policy type. NCon($M$) represents the normalized contribution of meteorology change during 2013–2017, which equals the product of the observational PM$_{2.5}$ balance (from 2013 to 2017) and the proportion of simulated meteorology contribution (in the simulated contributions of all factors). Similarly, NCon($M$) and NCon($pi$) represent the normalized contribution of regional emission reductions and each local control policy type.

## 3 Results and discussion

### 3.1 Observed changes in surface air quality in Beijing during 2013–2017

In 2013, air pollution was the major environmental problem in Beijing and its surrounding regions (Zhang et al., 2016). In addition to the severe and persistent haze events, the annual mean PM$_{2.5}$ concentration was 89.5 $\mu$g m$^{-3}$ in Beijing.

Furthermore, the concentrations of other major air pollutants were also at fairly high levels, with $56.0\,\mu g\,m^{-3}$ for $NO_2$, $26.5\,\mu g\,m^{-3}$ for $SO_2$, $183.4\,\mu g\,m^{-3}$ for $O_3$ (the statistic refers to the annual average of daily maximum 8 h sliding, 90th percentile concentration), and $3.4\,mg\,m^{-3}$ for CO (the statistic refers to the annual average of daily 24 h, 95th percentile concentration) (BMEP, 2014).

During 2013–2017, the annual average concentrations of $SO_2$, $NO_2$, $PM_{2.5}$, and $PM_{10}$ decreased steadily in Beijing (Fig. S3a). $SO_2$ had the most significant decrease rate of $-69.8\,\%$ (Fig. S3b), indicating the great effectiveness of the clean air actions on $SO_2$ emission control. $PM_{2.5}$ had the second greatest decrement of $35.2\,\%$, and the annual concentration of $PM_{2.5}$ in 2017 was $58\,\mu g\,m^{-3}$, overfulfilling the air quality targets in the APPCAP.

Although the annual average $PM_{2.5}$ concentrations decreased remarkably during 2013–2017, the monthly concentration varied substantially in different years, as shown in Fig. 3. Compared with the 2013 level, the average $PM_{2.5}$ concentrations of each month in 2017 all had a notable decline and presented a similar trend from July to December. However, compared with the 2016 level, the $PM_{2.5}$ pollution in January and February was more severe than in 2017, while it noticeably improved after October and decreased by nearly $66.3\,\%$ in December (from $130.7\,\mu g\,m^{-3}$ in 2016 to $44.0\,\mu g\,m^{-3}$ in 2017). The monthly $PM_{2.5}$ concentrations in November and December in 2016 were also much higher than those in 2013. A heavy $PM_{2.5}$ pollution episode occurred in the autumn of 2016 and the winter of 2016–2017. However, the $PM_{2.5}$ concentrations noticeably decreased after September in 2017 compared with both 2013 and 2016. The observed $PM_{2.5}$ trends indicate that the emission trend and intensity are major factors in the variation in $PM_{2.5}$ concentrations, while the meteorology changes also play an important role. The quantification of the contributions of emission control and meteorology changes will benefit numerous future applications.

### 3.2 Attribution of the 2013–2017 emission reduction in Beijing to specific measures

Based on the MEIC model and the detailed local bottom-up emission inventory, Beijing's atmospheric emissions were updated by year and by sector, as shown in Fig. 4. Furthermore, the attribution of emission reductions in Beijing to specific control measures during 2013–2017 and 2016–2017 was displayed in Fig. 5.

The major air pollutant emissions in Beijing in 2013 are estimated as follows: 95 kt of $SO_2$, 218 kt of $NO_x$, 273 kt of volatile organic compounds, and 81 kt of $PM_{2.5}$. The power and heating sector and the residential sector were the major sources of $SO_2$ emissions, accounting for $45.1\,\%$ and $40.6\,\%$, respectively. $NO_x$ emissions mainly came from mobile sources, which contributed $67.2\,\%$. Solvent use, mobile sources, and industry made notable contributions to VOC

emissions, accounting for $32.0\,\%$, $23.8\,\%$, and $23.7\,\%$, respectively. Fugitive dust and the residential sector were the major emitters of $PM_{2.5}$, with proportions of $48.7\,\%$ and $26.2\,\%$. However, the implementation of the Beijing Action Plan had a significant impact in terms of local emission reductions. Compared with 2013, Beijing's anthropogenic emissions in 2017 were estimated to have decreased by $83.6\,\%$ for $SO_2$, $42.9\,\%$ for $NO_x$, $42.4\,\%$ for VOCs, and $54.7\,\%$ for $PM_{2.5}$. Furthermore, the structure of the emission proportions also changed. For $NO_x$ emissions, transportation still remained the largest emitter of $NO_x$ in 2017 but represented a much higher proportion in 2017 than in 2013. The contributions of other sectors, especially power and heating, decreased. With notable contributions of VOC emission reductions in the residential and industrial sectors, the proportions of these two sectors noticeably decreased in 2017, and solvent use as well as transportation became the major emitters. For $PM_{2.5}$, through the effective measures implemented in the residential, industrial, and power and heating sectors, these sectors emitted less $PM_{2.5}$ in 2017 than in 2013, and the majority of $PM_{2.5}$ emissions came from fugitive dust.

In general, the power and heating, industry, and residential sectors exhibited the most notable emission reductions during 2013–2017. The variations in emissions by sector and year are mainly attributable to air pollution control policies and measures. As previously mentioned, seven types of air pollution control measures were simultaneously contributing to the emission reduction process. According to our research, during 2013–2017, *coal-fired boiler control* and *clean fuels in the residential sector* had the most notable effects on $SO_2$ emission reductions and reduced $SO_2$ emissions by 35 and 28 kt, respectively, accounting for $44.0\,\%$ and $35.2\,\%$ of the total (Fig. 6a; Table S5). Coal combustion was regarded as the major source of $SO_2$ emissions in Beijing, where coal was primarily used for residential heating and cooking, coal-fired boilers, and power plants. The great emission reduction in $SO_2$ indicated accurate source identification and effective emission control in the past 5 years. However, we should also notice that end-of-pipe controls on coal combustion in Beijing have been developed and almost finished recently, leaving little room for further emission reduction. Therefore, the adjustment and optimization of the energy structure would be the most effective and dominant pathway for mitigating coal combustion pollution in the future. According to several studies on the source apportionment of $PM_{2.5}$ in Beijing, the transportation sector accounted for a major part of $PM_{2.5}$ pollution in 2013, and its contribution has significantly increased since then (Li et al., 2015, 2017; Hua et al., 2018; Y. Zhang et al., 2018). *Vehicle emission control*, including both on-road and off-road vehicles, was the biggest contributor to $NO_x$ emission reductions with an estimated total reduction of 44 kt $NO_x$, accounting for $47\,\%$ of the total reductions (Fig. 6a; Table S5). *Improved end-of-pipe control* reduced the $NO_x$ emissions by 10 kt in total and accounted for $10.3\,\%$ of the total $NO_x$ reductions. In view of the widespread con-

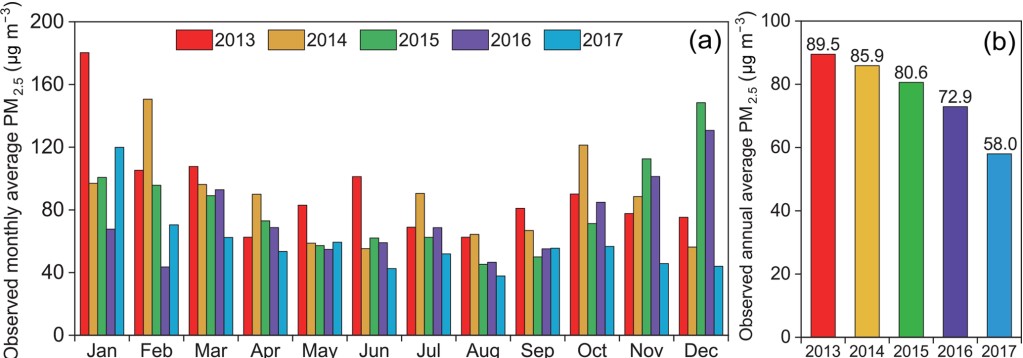

**Figure 3.** Observed monthly **(a)** and annual **(b)** average PM$_{2.5}$ concentrations in Beijing during 2013–2017.

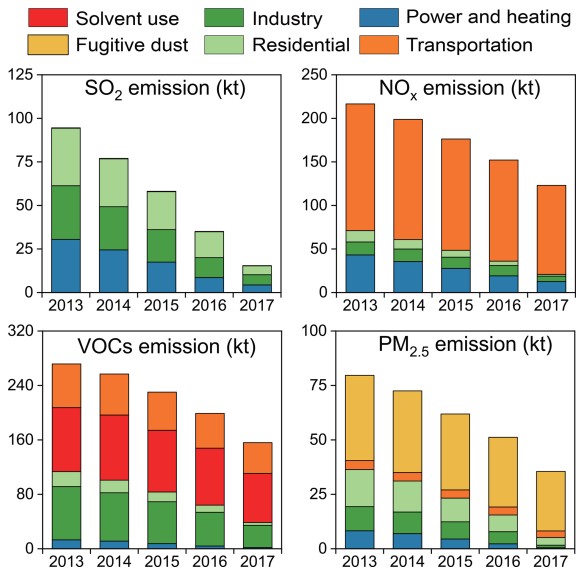

**Figure 4.** Changes in anthropogenic emissions of SO$_2$, NO$_x$, VOCs, and primary PM$_{2.5}$ in Beijing during 2013–2017.

version of combustion equipment from coal-based to oil/gas-based equipment, several measures were taken to improve the end-of-pipe control in response to the potential increase in NO$_x$ emissions, including the application of low-nitrogen-burning (LNB) technologies. A large number of gas-fired or oil-fired boilers, equivalent to 34 000 MV, have been renovated, decreasing NO$_x$ emissions by nearly 7.5 kt (Table S5). Benefitting from the advanced planning of VOC pollution control and scientific source apportionment, VOC control measures were also as effective as other pollutant control measures in Beijing during 2013–2017. *Integrated treatment of VOCs* had the most prominent achievement in reducing VOC emissions, with a reduction of 57 kt and a proportion of 49.3 % (Fig. 6a; Table S5). *Vehicle emission control* and *optimize industrial structure* also effectively reduced VOC emissions, accounting for 16.1 % and 11.4 %, respectively (Fig. 6a; Table S5). For PM$_{2.5}$ emission control, *clean fu-*

*els in the residential sector*, *fugitive dust control*, *coal-fired boiler control*, and *optimize industrial structure* all made noticeable contributions, which reduced the PM$_{2.5}$ emissions by 13, 11, 10, and 6 kt, respectively, and accounted for 90.3 % of the total (Fig. 6a; Table S5). In recent years, fugitive dust has gradually become the most dominant source of PM$_{2.5}$ emissions, but the relevant control measures are considered less effective than measures focused on coal combustion and the industry sector. Moreover, as the PM$_{2.5}$ emissions from the industry sector and coal combustion have gradually decreased and become better managed, fugitive dust, including road dust, construction dust, and stock dump dust, has become the most challenging target for future PM$_{2.5}$ emission control. In general, *coal-fired boiler control*, *clean fuels in the residential sector*, *optimize industrial structure*, and *vehicle emission control* made significant contributions to pollutant emission reductions in Beijing during 2013–2017 overall, while *integrated treatment of VOCs* and *fugitive dust control* achieved prominent reductions in VOCs and PM$_{2.5}$ emissions.

To ensure that the national air quality targets of the APP-CAP could be achieved as scheduled, Beijing implemented a series of stronger and more targeted pollution control policies and measures since 2016. For energy structure adjustment, measures associated with *clean fuels in the residential sector* were enhanced. A total of 92 000 households in urban areas and 369 000 households in rural areas converted the scattered coal-based fuels into clean fuels, close to the total amount of 2013–2016. For industrial structure adjustment, Beijing strengthened the elimination and management of small, cluttered, and heavily polluting factories. More than 6500 factories were eliminated during 2016–2017, approaching 1.5 times compared with the total amount during 2013–2016. During 2016–2017, SO$_2$, NO$_x$, VOCs, and PM$_{2.5}$ were estimated to have decreased by 19.6, 29.0, 42.9, and 15.7 kt, respectively (Table S5). *Clean fuels in the residential sector*, *coal-fired boiler control*, and *optimize industrial structure* were the top three most effective local measures during this period. In addition, with the enhanced management

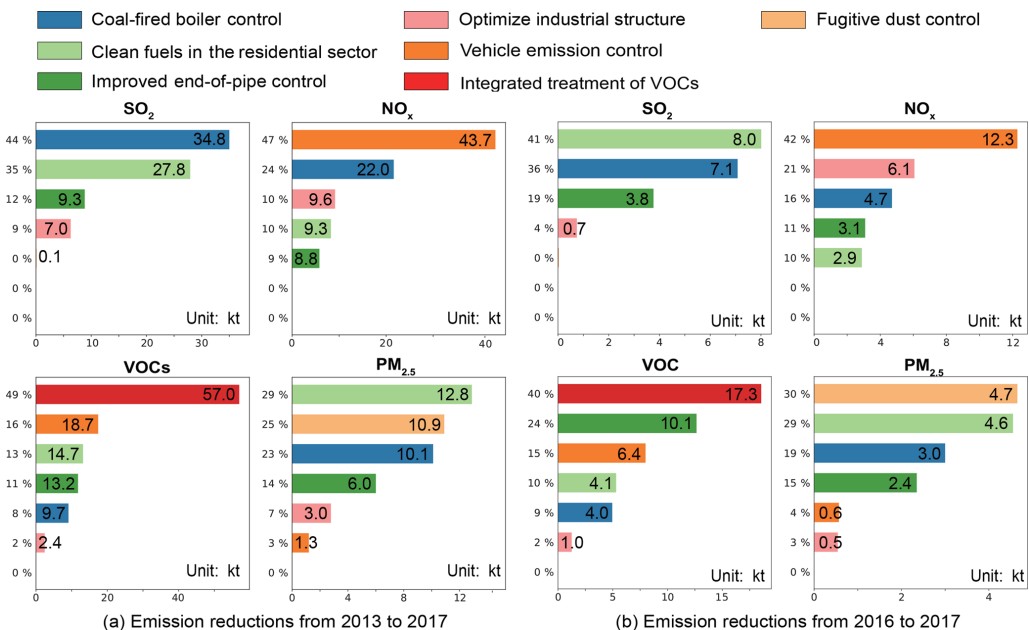

**Figure 5.** CE5 Emission reductions of $SO_2$, $NO_x$, VOCs, and primary $PM_{2.5}$ attributed to seven types of control policies in Beijing for the periods of 2013–2017 **(a)** and 2016–2017 **(b)**. The relative contribution of each policy to the total emission reduction is presented on the $Y$ axis. The number on each bar represents the absolute emission reductions by the relevant control policy.

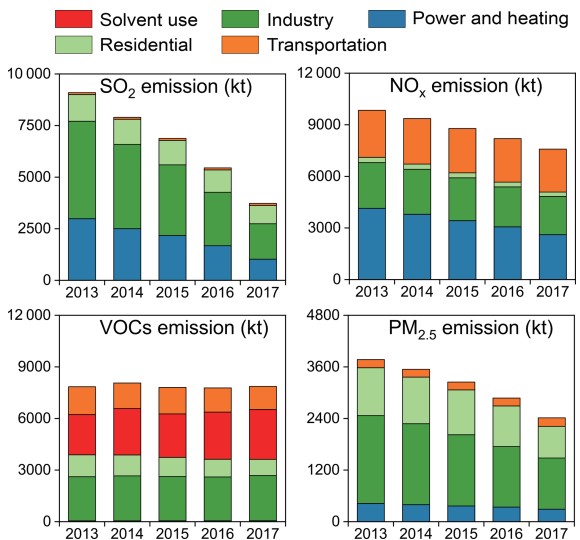

**Figure 6.** Changes in anthropogenic emissions of $SO_2$, $NO_x$, VOCs, and primary $PM_{2.5}$ in the areas surrounding Beijing during 2013–2017. The regions include Tianjin, Hebei, Henan, Shandong, Shanxi, and Inner Mongolia.

of non-point air pollution sources, including fugitive dust, heavily polluting vehicles, and domestic solvent use, the relevant control measures generated more remarkable emission reductions in this period than the previous periods. For instance, *fugitive dust control* was estimated to have decreased the $PM_{2.5}$ emissions by 4.7 kt during 2016–2017, represent-

ing 42.5 % of the total $PM_{2.5}$ emission reductions by *fugitive dust control* during 2013–2017.

## 3.3 Emission reduction in surrounding regions during 2013–2017

According to our previous research (Zheng et al., 2018b), the implementation of national clean air actions has brought conspicuous emission reductions in Beijing's surrounding regions (including Tianjin, Hebei, Henan, Shandong, Shanxi, and Inner Mongolia) from 2013 to 2017 (China State Council, 2018). Figure 6 showed the updated emission inventory of Beijing's surrounding regions by year and sector during 2013–2017.

According to Fig. 6, $SO_2$ and $PM_{2.5}$ emissions presented rapid decreasing trends from 2013 to 2017, while the trend of VOC emissions remained steady with a slight increment. Compared with 2013, $SO_2$, $NO_x$, and $PM_{2.5}$ emissions were estimated to decrease by 59.5 %, 22.9 %, and 36.6 %, respectively, while VOC emissions increased slightly by 0.2 % (Zheng et al., 2018b). During 2013–2017, the industry and the power and heating sectors made the most prominent contributions to $SO_2$ emission reductions, which decreased $SO_2$ emissions by 3021 and 2000 kt, respectively, indicating that the $SO_2$ emissions control measures were quite effective (Cai et al., 2017; Shao et al., 2018). Reductions in the $NO_x$ emissions mainly came from the power and heating sector, with a reduction of 1541 kt within 5 years. For $PM_{2.5}$ emission reductions, industrial sectors were the greatest contributors, with reductions of 621 and 528 kt. The VOC emissions in the

surrounding regions continued to increase, especially in the solvent use sectors. During the process of implementing national clean air actions, the surrounding regions also carried out several valid measures to control VOC emissions, such as the supervision and repair of gasoline stations, oil tankers, and oil transfer processes and the integrated treatment and management of petrochemical and refinery industries (Hui et al., 2019). The VOC emissions from the residential and transportation sectors in 2017 decreased by 20.7 %. However, due to the lack of thorough regulation of chemical industries and the ineffective end-of-pipe control of solvent use sources, the total VOC emissions increased slightly in 2017 (Zheng et al., 2018b).

### 3.4 Decomposition of PM$_{2.5}$ concentration changes in Beijing during 2013–2017

#### 3.4.1 Modelled PM$_{2.5}$ air quality changes in Beijing during 2013–2017

According to the base simulation results, the annual average PM$_{2.5}$ concentrations of Beijing decreased by 32.8 µg m$^{-3}$ from 2013 to 2017 and by 16.6 µg m$^{-3}$ from 2016 to 2017, which agrees well with the observed decreases (31.5 µg m$^{-3}$ from 2013 to 2017 and 14.9 µg m$^{-3}$ from 2016 to 2017). Although there was a steady decline in PM$_{2.5}$ concentrations of Beijing during 2013–2017, the trends of PM$_{2.5}$ compositions varied differently. The simulation results of base cases (which adopted the real meteorology and emissions of each year) showed that the sulfate (SO$_4^{2-}$) and organic matter (OM) were the dominant species for the decline in PM$_{2.5}$ concentrations during 2013–2017, with the decrement of 7.5 µg m$^{-3}$ (56.6 %) and 9.6 µg m$^{-3}$ (40.5 %), respectively. The contribution of SO$_4^{2-}$ to the total PM$_{2.5}$ also noticeably decreased, from 15.3 % in 2013 to 10.7 % in 2017; and OM proportion decreased from 27.5 % in 2013 to 26.5 % in 2017. The rapid decrement of SO$_4^{2-}$ was consistent with the remarkable SO$_2$ emission reductions in Beijing during 2013–2017. Along with the effective SO$_2$ emission control measures, SO$_4^{2-}$ was basically no longer the key contributor leading to heavy pollution in Beijing, while the nitrate-driven haze pollution has become more dominant in Beijing in recent years, especially in the summertime (Li et al., 2018). The decrement of OM was mainly caused by the prominent emission reductions of primary organic carbon (mainly from residential burning and other coal combustion sources). VOC emission reductions also contributed to the OM decreasing; however, due to the insufficient simulation of SOA formulations in the CMAQ model, the contributions of VOC emission control might be underestimated. In contrast, nitrate (NO$_3^-$) increased in 2014–2016 and kept basically the same concentration level in 2017 (10.4 µg m$^{-3}$) as in 2013 (10.9 µg m$^{-3}$). However, the contribution of NO$_3^-$ to the total PM$_{2.5}$ increased a lot, from 12.7 % in 2013 to 19.4 % in 2017.

The specific concentration and proportion trends of PM$_{2.5}$ concentrations can be found in Table S6.

Figure 7 shows the spatial distribution of PM$_{2.5}$ concentrations in Beijing and the surrounding areas in 2013, 2016, and 2017 (panels a–c), along with the total PM$_{2.5}$ changes and the changes due to major contributing factors from 2013 to 2017 (panels d–f) and from 2016 to 2017 (panels g–i). In 2013, some typical regions, such as southern Beijing and most of the cities of Tianjin, Tangshan, Baoding, Shijiazhuang, Handan, and Anyang, suffered intense PM$_{2.5}$ pollution. After implementing the APPCAP and local air pollution control policies, severe pollution was mitigated in most regions, although several heavily polluted spots still existed. However, Beijing had successfully removed itself from the list of heavily polluted areas. According to the base simulation results (Fig. 7a–c), Beijing, especially the southern area, had the most notable PM$_{2.5}$ decrease among all parts of the third nested simulation domain. The municipality of Tianjin and the southwestern Hebei province also achieved prominent abatements in PM$_{2.5}$ pollution. Based on the spatial distributions of total PM$_{2.5}$ changes and changes due to major contributing factors in Beijing (Fig. 7d–i), the control of emissions dominated the PM$_{2.5}$ changes in both 2013–2017 and 2016–2017; however, the favourable effects of meteorological changes during 2016–2017 were much more remarkable. We further estimated and quantified the contributions of each factor as follows.

#### 3.4.2 Contribution from changes in meteorological conditions

According to the simulation results of the base cases and fixed-emission sensitivity experiments (Tables 2, S6), the meteorological conditions in 2017 were found to be more favourable than those in the previous periods, especially 2016. During 2013–2017, changes in meteorological conditions contributed 3.8 µg m$^{-3}$ to the PM$_{2.5}$ air quality improvements, accounting for 12.1 % of the total abatements. Under the meteorological conditions of 2013 and the emission level of 2017, the annual average PM$_{2.5}$ concentration of Beijing would have decreased from 90 to 62–62.5 µg m$^{-3}$ and would not have achieved the air quality targets established in the APPCAP. During 2016–2017, the favourable effects of meteorology changes became much more striking and contributed 4.4 µg m$^{-3}$, accounting for 29.5 % of the total PM$_{2.5}$ abatements from 2016 to 2017. Similarly, under the meteorological conditions of 2016, the PM$_{2.5}$ level in Beijing in 2017 would have decreased to 62.5–63.0 µg m$^{-3}$, still in excess of the APPCAP target.

From the perspective of annual average analysis, changes in meteorology generally had a beneficial effect on air pollution mitigation in 2017; however, the impact varied greatly in the monthly analysis. Figure 8 showed the monthly average simulated PM$_{2.5}$ concentrations in the two fixed-emission sensitivity experiments. Compared with the meteorological

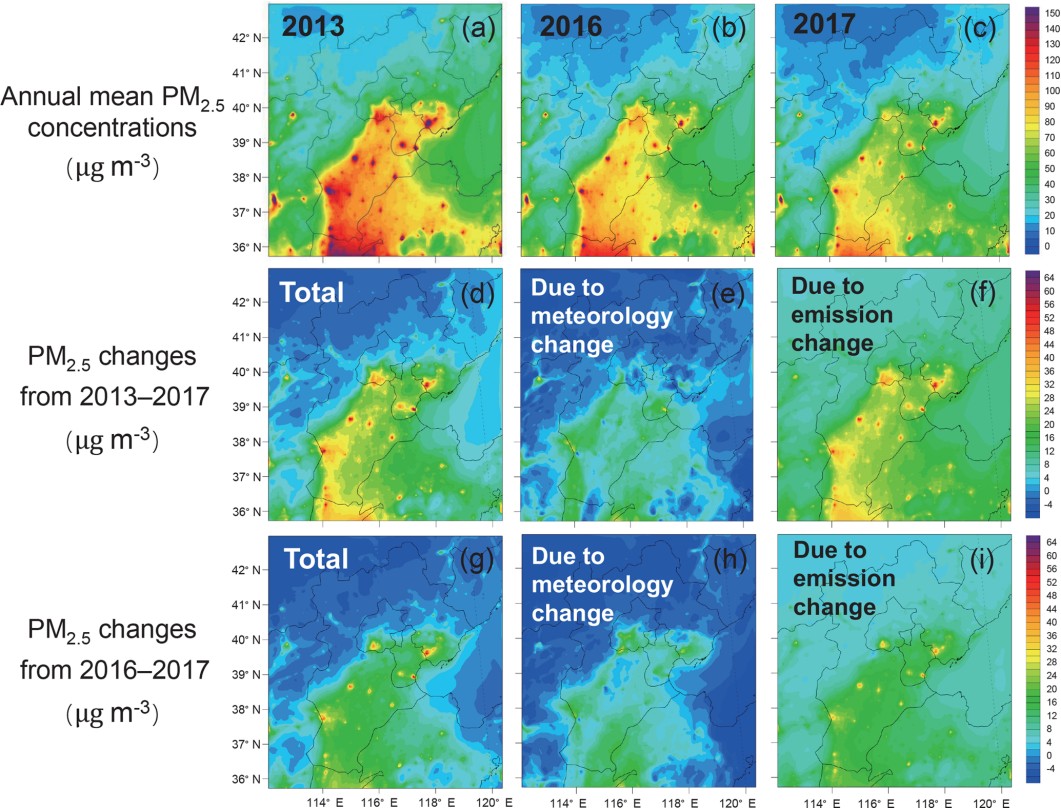

**Figure 7.** Changes in CMAQ-simulated annual mean $PM_{2.5}$ concentrations. **(a–c)** Base simulations of 2013, 2016, and 2017; **(d–f)** total $PM_{2.5}$ changes and the changes attributed to meteorology and emission variations during 2013–2017; **(g–i)** total $PM_{2.5}$ changes and the changes attributed to meteorology and emission variations during 2016–2017.

conditions of 2013 (Fig. 8a), the meteorological conditions of 2017 became better in winter, especially in January and February. Under the anthropogenic emissions of 2017, the meteorological conditions of January and February 2013 would have increased the $PM_{2.5}$ concentration by 22.5 % and 37.7 %, respectively. However, the meteorological conditions of 2017 were worse than those of 2013 in spring and summer, especially in April, May, and July. The air quality was good in the first few months of 2016, and the simulation results also indicated that the meteorological conditions during this period in 2017 were much worse than those in 2016, increasing the $PM_{2.5}$ concentrations by 51.6 $\mu g\,m^{-3}$ in January and 28.6 $\mu g\,m^{-3}$ in February (Fig. 8b). However, the conditions improved in the following months, especially during October to December. If the meteorological conditions remained the same as those in 2016, the monthly average $PM_{2.5}$ concentrations of October, November, and December would have increased by 25.4 %, 58.0 %, and 92.4 %, respectively. It is worth noting that severe $PM_{2.5}$ pollution and haze events always occur in winter in North China; therefore, remarkable improvements in the meteorological conditions in January, February, November, and December would contribute greatly to the mitigation of annual $PM_{2.5}$ concentrations.

### 3.4.3 Contribution from local and regional emission reduction measures

Although the changes in meteorological conditions were favourable for $PM_{2.5}$-related air quality improvements in Beijing, the control of emissions was still the dominant factor in $PM_{2.5}$ abatement in recent years and contributed the reductions of 27.7 $\mu g\,m^{-3}$ (accounting for 87.9 %) and 10.5 $\mu g\,m^{-3}$ (accounting for 70.5 %) in 2013–2017 and 2016–2017, respectively.

According to the simulation results of the regional fixed-emission sensitivity experiments (Tables 2 and S6), the contributions of regional emission reductions to the $PM_{2.5}$ abatements in Beijing were 7.1 $\mu g\,m^{-3}$ during 2013–2017 and 2.5 $\mu g\,m^{-3}$ during 2016–2017, accounting for 22.5 % and 16.8 %, respectively. The results indicated that, by implementing the APPCAP, regional provinces and cities around Beijing achieved notable emission control effects. In particular, emission reductions in the industry and power sectors have made striking contributions to $PM_{2.5}$-related air quality improvements in regional areas.

In addition to the impacts of meteorology changes and regional emission reductions, the contributions of local emission control to the $PM_{2.5}$ abatements in Beijing were es-

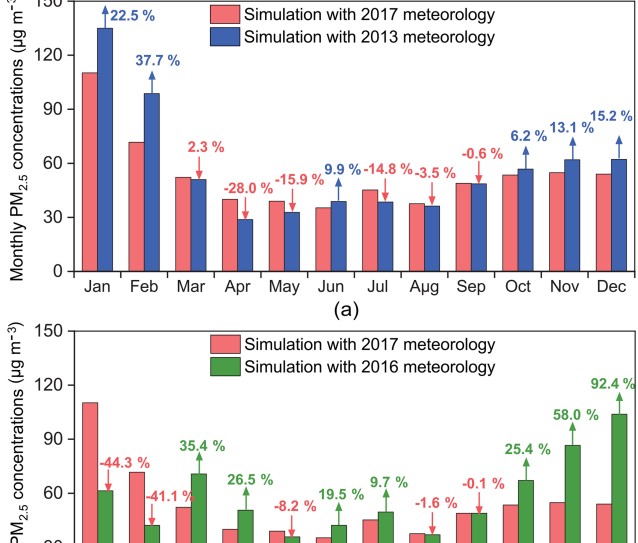

**Figure 8.** CMAQ-simulated monthly PM$_{2.5}$ concentrations in Beijing under different meteorological conditions. The numbers shown in each panel represent the monthly relative change rates of fixed-emission simulation results compared with the base simulation results.

timated to be $20.6\,\mu g\,m^{-3}$ (2013–2017) and $8.0\,\mu g\,m^{-3}$ (2016–2017), accounting for 65.4 % and 53.7 %, respectively. According to the results of the measure-related sensitivity experiments (Table S6), we further decomposed the contributions due to local emission control into each specific measure. As Fig. 9 shows, during 2013–2017, *coal-fired boiler control* made the largest contribution of $5.9\,\mu g\,m^{-3}$, accounting for 18.7 % of the total decrement. *Clean fuels in the residential sector* was the second greatest contributor after *coal-fired boiler control*, decreasing PM$_{2.5}$ concentrations by $5.3\,\mu g\,m^{-3}$. Measures associated with *optimize industrial structure* also effectively reduced PM$_{2.5}$ concentrations, with a decrease of $3.2\,\mu g\,m^{-3}$ and a proportion of 10.2 %. Measures associated with *fugitive dust control*, *vehicle emission control*, *improved end-of-pipe control*, and *integrated treatment of VOCs* had relatively minor contributions and reduced the PM$_{2.5}$ concentrations in Beijing by 7.3 %, 6.0 %, 5.7 %, and 0.6 %, respectively, from 2013 to 2017 (Fig. 9a). During 2016–2017, *clean fuels in the residential sector*, *coal-fired boiler control*, and *optimize industrial structure* were the top three contributors to the PM$_{2.5}$ abatements among all local policies, accounting for 14.1 %, 12.8 %, and 10.1 % of the total (Fig. 9b). These results highlight the great enhancement in the control of bulk coal use and the elimination of small, clustered, and heavily polluting factories during this period.

In summary, the improvement in the PM$_{2.5}$-related air quality in Beijing was decomposed, and the results are shown in Fig. 10. During 2013–2017, meteorology changes, surrounding emission reductions, and local emission control contributed 3.8, 7.1, and $20.6\,\mu g\,m^{-3}$, respectively, accounting for 12.1 %, 22.5 %, and 65.4 %. *Coal-fired boiler control*, *clean fuels in the residential sector*, and *optimize industrial structure* were the top three contributors among all local emission control policies. Emission reduction was the most dominant factor in the air quality improvements in Beijing during this period. For 2016–2017, the contributions of meteorology changes, surrounding emission reductions, and local emission control were 4.4, 2.5, and $8.0\,\mu g\,m^{-3}$, respectively. The favourable meteorological conditions during this period had a remarkable effect, accounting for 29.5 % of the total pollution reduction. The top three local control measures of this period were same as those of 2013–2017 but had a different order. *Clean fuels in the residential sector* and *optimize industrial structure* made larger contributions.

### 3.5 Uncertainties and limitations

We built a model-based decomposition framework and attributed the PM$_{2.5}$ abatements in Beijing during 2013–2017 and 2016–2017; however, certain uncertainties and limitations remain. The major uncertainties and limitations of this work are discussed below.

#### 3.5.1 Uncertainties of the zero-out approach

Although various methods have been developed to quantify the source of PM$_{2.5}$ and evaluate their contributions, such as receptor-based methods (like CMB and PMF), trajectory-based methods (like PSCF and EEI), and source-oriented methods (like CAMx-PSAT and CMAQ-ISAM) (Li et al., 2015), they can hardly consider the meteorology and emission changes simultaneously. Therefore, the zero-out approach might be a better choice to attribute the contribution of local and regional emission control as well as meteorology changes under one complete decomposition framework. The zero-out method is also widely used in estimating the contribution of air pollution sources (Lelieveld et al., 2015; Han et al., 2016; Baker et al., 2016; Q. Zhang et al., 2017; R. Zhang et al., 2017; Ni et al., 2018).

However, the response of PM$_{2.5}$ formulation is not linear to the meteorology and emission changes; thus, the zero-out approach would introduce extra bias in research. The nonlinear effects of the analysis period of 2013–2017 could be evaluated by the following equation (Q. Zhang et al., 2017).

$$\text{Bias} = (\text{SCon}\,(M) + \text{SCon}\,(S) + \sum_{i=1}^{7}\text{SCon}\,(pi)) \\ - (\text{SPM}_{2.5}\,(E_{L13S13}M_{13}) - \text{SPM}_{2.5}\,(E_{L17S17}M_{17})), \quad (7)$$

where $\text{SPM}_{2.5}\,(E_{L13S13}M_{13})$ and $\text{SPM}_{2.5}\,(E_{L17S17}M_{17})$ represent the direct simulated PM$_{2.5}$ concentration of the base case in 2013 and 2017. The balance of their values is the actual PM$_{2.5}$ decrement during 2013–2017 under the mixed

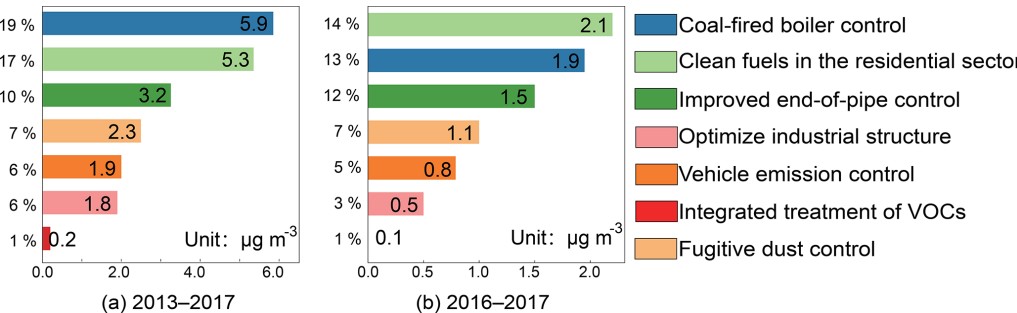

**Figure 9.** Contributions of the seven types of control policies to the $PM_{2.5}$ abatements in Beijing for the periods of 2013–2017 **(a)** and 2016–2017 **(b)**. The relative contribution rate of each policy to the total $PM_{2.5}$ abatements is presented on the $Y$ axis. The number on each bar represents the normalized absolute $PM_{2.5}$ abatement by the relevant control policy.

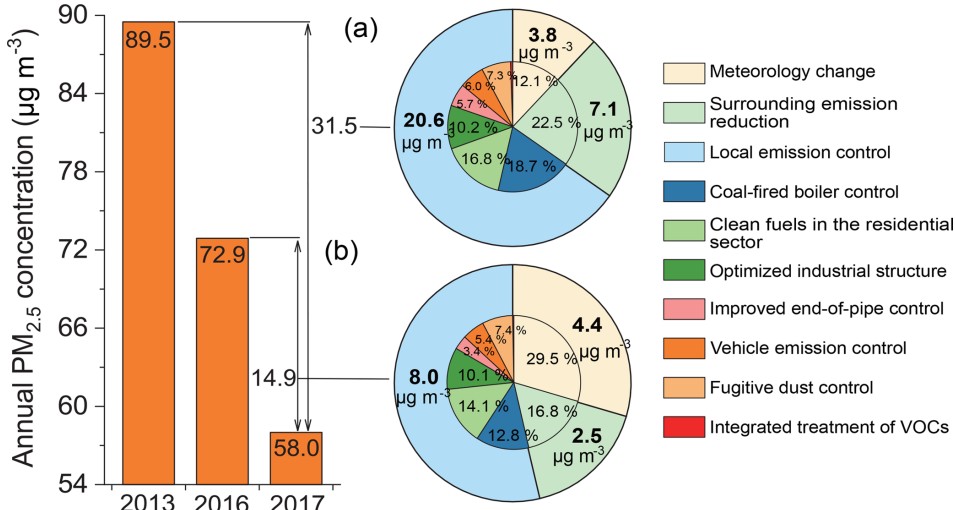

**Figure 10.** Decomposition of improved $PM_{2.5}$ air quality in Beijing during **(a)** 2013–2017 and **(b)** 2016–2017.

impacts of meteorology change as well as regional and local emission reductions. The sum of SCon($M$), SCon($S$), and $\sum_{i=1}^{7}$ SCon($pi$) represents a linear result of all contributors during this period. The extra bias can be estimated as the difference between the linear addition and the actual decrement. According to Eq. (7), we estimated biases in the analysis of 2013–2017 were $1.4\,\mu g\,m^{-3}$, accounting for 4.3 %. Similarly, the absolute and relative biases in the analysis of 2016–2017 were estimated as $-0.6\,\mu g\,m^{-3}$ and $-3.6\%$. Both indicated the non-linear effects are relatively small and acceptable.

### 3.5.2   Uncertainties in emission estimates

The incomplete research and investigation of activity rates, emission factors, and removal efficiencies would introduce uncertainties in estimating the emission trends and magnitudes, as well as the measure-based emission reductions.

Based on our previous work, the uncertainty of MEIC was estimated to be $\pm 12\%$ for $SO_2$, $\pm 31\%$ for $NO_x$, $\pm 70\%$ for CO, $\pm 68\%$ for VOCs, $\pm 130\%$ for $PM_{2.5}$, $\pm 208\%$ for BC, and $\pm 258\%$ for organic carbon (OC) (Zhang et al., 2009;

Zheng et al., 2018a). The larger uncertainty of BC and OC is mostly because their major emitters are much more scattered and harder to investigate or quantify, such as heavy diesel vehicles and residential burning. The uncertainty of the updated emission inventory of China during 2010–2017 was evaluated in Zheng's work by comparing the emissions with observations (Zheng et al., 2018b) and reported a good agreement.

Similarly, we discussed the uncertainty of emission trends and relative change ratios in BJ-EI. Ground-based, satellite-based observational data and the estimated emissions form BJ-EI were all normalized to the base year, 2013. We compared the observational concentration trends of major pollutants with their precursor emissions (Fig. S5). A good agreement was found in the trends of $SO_2$ emission, OMI $SO_2$ column, and surface observational $SO_2$ concentration, with a decreasing ratio of 83.6 %, 77.6 %, and 69.8 %, respectively (Fig. S5a), indicating a relatively small uncertainty in $SO_2$ emission estimation. The surface concentration trend during 2016–2017 became flatter than the OMI $SO_2$ columns trend.

This is partly because during this period the surface $SO_2$ emission became gradually steady, while the high-stack $SO_2$ emission reductions became more significant, especially in regional areas. The $NO_x$ emission trend was basically consistent with the variation in $NO_2$ tropospheric vertical column, decreasing 43.2 % and 40.3 %, respectively, but both were lower than the surface $NO_2$ concentration trend (Fig. S5b). This phenomenon might be caused by the meteorology impacts (Uno et al., 2007), chemical reactions of nitrous oxides (Valin et al., 2011), and the overestimation of surface $NO_2$ observations (Lamsal et al., 2010). Satellite-derived and surface observational $PM_{2.5}$ concentrations decreased 36.0 % and 35.2 % respectively during 2013–2017 in Beijing, and both agreed with the trends of primary $PM_{2.5}$ and precursor emissions (Fig. S5c). Among all precursors, the decreasing rate of $SO_2$ was more significant and rapid than that of $NO_x$ and $HN_3$, which was consistent with the simulation results in which the proportion of sulfate in $PM_{2.5}$ noticeably decreased while the contributions of nitrate and ammonium increased (Fig. S4). The decrement of VOC emissions contributed to $PM_{2.5}$ abatements by decreasing OM. A similar phenomenon was also reported in previous research (Shao et al., 2018). In general, the relatively good coherence of emission and observation variations indicated that the BJ-EI basically well quantified the actual emission trends and variations in Beijing during 2013–2017.

Estimation of measure-based emission reductions is another major aspect of introducing uncertainty. However, the uncertainty of this aspect is hard to quantify. Emission control measures can have independent or integrated impacts on activity rate, emission factor, technology evolution, and end-of-pipe removal efficiency, which are all sensitive to emission calculation and remaining large uncertainty. *Coal-fired boiler control* was the most explicit control policy to quantify, for the detailed and accurate information of unit-based power plants and facility-based boilers. The amount of eliminated coal, newly increased alternative clean energy, the evolution of emission factors, and the removal efficiency of each boiler were collected in sufficient amounts, which can largely lower the uncertainty of the reduction estimation. Similar to *coal-fired boiler control*, *improved end-of-pipe control* mainly focused on the heavily polluted manufacturing factories and gas/oil-fired boilers. The facility-based information and the accurate eliminated capacities make the uncertainty relatively small. As for the policy of *clean fuels in the residential sector*, which reduced residential coal use, the eliminated traditional biofuels were collected at the county level; thus the uncertainty of activity rate estimation was relatively small. However, along with the promotion of coal quality (such as the lower sulfur and ash content) and the evolution of domestic burning equipment, the improvement of emission factors was hard to estimate, especially in the rural areas, which would introduce large uncertainty. Major limitations of estimating *optimize industrial structure* reductions came from the elimination of small, clustered, and polluting

factories. Different from highly polluted enterprises, the specific information of these scattered factories was hard to investigate. Although we knew a total of 11 000 such factories were phased out during 2013–2017, the activity rate, emission factor, or end-of-pipe control information were ambiguous, which would easily lead to the underestimation or overestimation of reductions from this sub-measure. *Integrated treatment of VOCs* included the VOC control of the chemical industry and solvent use sectors. The Yanshan company is the only chemical factory in Beijing, and the specific information of this factory made the estimation more reasonable. However, investigations for the amounts of solvent used were limited; meanwhile, the lack of emission factor measurements for various solvent-use-related sources also introduced large uncertainty (Li et al., 2017a). Reductions from *fugitive dust control* might be the most difficult one to estimate, with a larger uncertainty. This policy type contained the control of road dust and construction dust. Due to the lack of measurements for the real-time traffic flow, threshold friction velocities, surface roughness length, the efficiency of the road cleaning process, and other key parameters for emission calculation, we estimated the emission reductions from this sub-measure by the improvements of the cleaning process adopted ratio and the various road areas, which might not totally reflect the actual emission change of road dust. For the construction dust estimation, although we collected the information of each construction site and stock yard in Beijing, the indefinite emission process and factors would also create uncertainties for estimation. In general, emission reductions from the policies which focused on non-point and scattered emission sources, such as road dust, small and clustered factories, and various solvent use sources, are more difficult to quantify and would cause larger uncertainties. Additional detailed information and real-world measurements might help to lower these uncertainties.

### 3.5.3 Other uncertainties

The decomposed analysis was also affected by the inherent WRF-CMAQ uncertainties, as Sect. 2.4.2 discussed. The missing mechanism and insufficient simulation of SOA formulations in the CMAQ model might lead to the underestimation of the OM, as well as the contribution of VOC emission control. The rough vertical layers, the underestimation of nature source emissions, the defect of upper boundary simulation in the regional model, and the uncertainties of VOC emission inventories might all lead to the deficiency of $O_3$ underestimation. The lack of construction dust and the underestimation of the in-line windblown dust model might result in the lower $PM_{10}$ simulation in other regions. And the open biomass burning was not included in this study, which would also introduce certain uncertainties. Given that we focused more on the decomposed attribution of anthropogenic emission changes and meteorology impacts, and the simulations of $PM_{2.5}$ concentrations and compositions basically

captured the temporal and spatial variations, the uncertainties originated from this aspect were relatively small. However, further studies and efforts should made to improve the model simulations.

## 4   Concluding remarks

The remarkable decreases in the annual average $PM_{2.5}$ concentrations in Beijing from 2013 to 2017 and from 2016 to 2017 were the combined results of various factors. In this study, based on a series of numerical simulation experiments and a decomposed attribution analysis, local air pollution control policies, surrounding emission reductions, and favourable meteorological conditions were estimated to contribute 65.4 % (20.6 µg m$^{-3}$), 22.5 % (7.1 µg m$^{-3}$), and 12.1 % (3.8 µg m$^{-3}$), respectively, of the total $PM_{2.5}$ abatements in Beijing (31.5 µg m$^{-3}$) from 2013 to 2017 and 53.7 % (8.0 µg m$^{-3}$), 16.8 % (2.5 µg m$^{-3}$), and 29.5 % (4.4 µg m$^{-3}$), respectively, of the total $PM_{2.5}$ abatements (14.9 µg m$^{-3}$) from 2016 to 2017. During 2013–2017, air pollution control policies had the most dominant effect on $PM_{2.5}$ abatements, accounting for nearly 88 %, but the meteorological impacts have been considerable since 2016, especially in the winter of 2016–2017 and the autumn of 2017.

Under the Beijing Action Plan, anthropogenic emissions were reduced by 83.6 % for $SO_2$, 42.9 % for $NO_x$, 42.4 % for VOCs, and 54.7 % for $PM_{2.5}$ compared with the 2013 level. Under the APPCAP, the areas surrounding Beijing also reduced their pollutant emissions by 59.5 % for $SO_2$, 22.9 % for $NO_x$, and 36.6 % for $PM_{2.5}$. A measure-by-measure analysis showed that *coal-fired boiler control, clean fuels in the residential sector*, and *optimize industrial structure* were the most effective control measures in general for Beijing during 2013–2017 and 2016–2017, both in terms of emission reductions and $PM_{2.5}$ pollution mitigation.

The results indicated several options for future air pollution control in Beijing. The most notable effect of the Beijing Action Plan mainly came from the control of combustion, which suggests that power plants, coal-fired boilers, and residential burning have accounted for the majority of the air pollution sources for a long time. Consequently, Beijing should continue to optimize the city's energy structure to achieve a qualitative improvement in energy consumption. However, with the progress on air pollution control in Beijing, the contributions of combustion and industry emission control, of which the sectors and sources are relatively easy to identify and manage, have gradually decreased, and there is less room for further improvement. Pollutant emissions from domestic living, such as transportation, restaurant fumes, and residential solvent use, have increasingly accounted for larger proportions. Vehicles, VOC emission sources, and fugitive dust have gradually become the major and most difficult challenges for Beijing's future air pollution control. On the one hand, the government should further apply stronger and more effective management of non-point pollution sources arising from the demands for city development, such as catering enterprises, vehicles, off-road transportation, construction sites, and the use of solvents and coatings. More resources and investment, more accurate identification, and refined management strategies are needed for these diffuse pollution sources. On the other hand, the support and innovation of science and technology should be enhanced further, including not only high-technology strategies of pollutant removal and equipment renovation but also the understanding of pollution mechanisms and the identification of pollution sources. For instance, the scientific source apportionment of atmospheric particulates, the dynamic update of emission inventories, the application of widespread observation systems, the construction of pollution forecasts and warning systems, etc. should be developed further. A support system for air quality analysis, decision-making, implementation, assessment, and optimization should be established in the future to make qualitative leaps in environmental protection.

*Data availability.* Data used in this paper are available upon request from corresponding author (qiangzhang@tsinghua.edu.cn).

*Supplement.* The supplement related to this article is available online at: https://doi.org/10.5194/acp-19-1-2019-supplement.

*Author contributions.* QZ, JL, and KH conceived the study; JS, TC, and FS developed the bottom-up emission inventory over Beijing; XL, XD, and YY collected pollution control policies over Beijing; DT and YZ estimated regional emission reductions; YL collected and handled the satellite data; JC estimated emission reductions over Beijing and performed CMAQ experiments; JC and QZ prepared the paper with contributions from all co-authors.

*Competing interests.* The authors declare that they have no conflict of interest.

*Acknowledgements.* We appreciate Junlin Wang and her team at the Beijing Municipal Research Institute of Environmental Protection for their efforts in providing and summarizing Beijing's recent air pollution control policies. Any opinions expressed in this paper are the views of the authors and are not the views of the Beijing Municipal Bureau of Ecology and Environment or the Beijing Municipal Environmental Monitoring Center. We are also grateful for the comments of Itsushi Uno and other reviewers as well as for their valuable suggestions that helped us to improve the paper. CE6

*Financial support.* This research has been supported by the Beijing Municipal Environmental Protection Bureau and the National Natural Science Foundation of China (grant nos. 41571130032, 41571130035, and 41625020).

*Review statement.* This paper was edited by Hang Su and reviewed by Itsushi Uno and one anonymous referee.

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

**Remarks from the language copy-editor**

**Remarks from the typesetter**