# Peer review of "Dominant role of emission reduction in PM2.5 air quality improvement in Beijing during 2013-2017: a model-based decomposition analysis"

_Atmospheric Chemistry and Physics, 2018_

## Referee Comment (RC1) · Anonymous Referee #1 · 8 Jan 2019

Dominant role of emission reduction in PM2.5 air quality improvement
in Beijing during 2013-2017: a model-based decomposition analysis
by Jing Cheng et al.

General Comments

The main purpose of this paper is the discussion of the reasons of recent rapid PM2.5 decrease of Beijing, mainly by using the meteorological and emission sensitivities by chemical transport model. Based on several sensitivity simulations, this paper made a decomposition analysis framework to evaluate the impacts of local control policies, surrounding emission reductions and the meteorological changes on PM2.5 abatement in Beijing during 2013-2017 and 2016-2017. This paper made the most of important sensitivity analysis and explains the relative contribution of meteorology (12%), local emission (65%) and regional emission (23%) for the reduction of PM2.5 between 2017 and 2013. The results of detailed sensitivity analyses are useful for the understanding of PM2.5 reduction and environmental policy. Their results are very much reasonable and important. However it is difficult to find scientific uniqueness of this paper. The current version of manuscript could be published as the Technical Note. It is necessary to add the more scientific discussion in order to be accepted as a research paper.

Specific Comments

1) The paper mainly discussed the decreases of annual average PM2.5 concentration. However as shown in Figure 4, the authors are specifying the decrease of chemical compositions (i.e., SO2, NOx), it is necessary to compare the aerosol chemical composition changes too. The under estimation of sulfate in winter are reported in many previous papers, the comparison of sulfate between observation and model results are very much necessary.

2) Another important point is the OC (we found that model result are usually under estimated), so the detailed examinations of model reproductively of OC are necessary in order to discuss the emission change in VOC.

3) The authors show the very good agreement of PM2.5 reduction between observation and model results. It is usually difficult to have such good agreement. It is necessary to discuss the detailed reasons why model results are so good agreement from the view point of emission inventory, WRF model, model horizontal resolution, and CMAQ itself performance.

4) Zero-out emission sensitivity is used in this study by assuming the linearity. PM2.5 formation is usually nonlinear, so it is necessary why the authors are using zero-out method.

5) Section 2.3: Although we can follow by Zheng et al. (2018) to understand the emission inventory, it should be noted what is the major "updated". Especially, the inventory is named as "MEIC"; however, we noticed the emission amounts described in Zheng et al. (2018) and Li, M. (2017) are different. Does this mean "updated"? Taking into account the importance of emission inventory, more careful explanations and descriptions are needed here for the traceability of this kind of study.

6) Section 2.4.1: Model descriptions are insufficient. Model calculation was conducted after only 10 days spin-up. How does the 3 years WRF simulation perform?  i.e., with or without FDDA? Need more detailed descriptions.

7) Does CMAQ study include the Asian dust? If yes, need discussion of model accuracy and problems.

8) Emission sensitivity study was conducted by each emission sector base. It is necessary to include the discussion of the accuracy (or error bars) of

emission estimate for each sector base.

9) Section 2.5 is unclear. The description of model sensitivity has to be rewrite. Equations of (1) – (3) are unclear. I think Equations of (2) and (3) are not NORMALIZED RESUTS.

10) Section 3.2 Without the model evaluation from 2013 to 2017, it is hard to discuss the source attribution results by model. From the current manuscript, we can only find time-series on 2013, 2016, and 2017 for PM2.5 and statistic evaluation only on 2017. The model evaluation is inadequate at the current manuscript. For example, Figure 3 and Figure S3 can be presented in the same form for model. See also my minor comments 4) for O3 performance.

11) The results shown here should be interpreted in depth. On 2013, especially the peak on January, model sometimes overestimated observed PM2.5. However, model simulated same level or sometimes underestimated high concentrations during winter on 2016 and 2017. Actually, the model negative-bias is larger in 2017 compared to 2013 (Table 2). Therefore, the source attribution results based on scenario analysis adopted in this study can be strongly reflected by emission variation rather than the observed facts.

12) Figures 5 and 9 are unfriendly. It need more detailed explanations or improve the presentation of figures. The basic information is same as Figure 10, so it might be better to modify Figures 5 and 9 into Figure 10 format.

13) Section 3.3: The discussion in this section needs relevant references.

Minor Comments

1) It have been reported that WRF should be updated version 3.9.0.1 or later for the upgraded NCEP dataset after 12 UTC, 19 July 2017.
   http://www2.mmm.ucar.edu/wrf/users/wpsv3.9/known-prob-3.9.html
   The used version is 3.8, but how did the authors solve this problem?

2) What is the horizontal grid resolution in second domain?

3) What is the lateral boundary condition for first domain? It will be taken from global chemical transport model, but did the global model consider year-to-year emission variation? If not, how can we conclude the importance of global-scale impacts on the air quality in China?

4) Considering the current modeling application over East Asia, the vertical 14 layers from surface to 10 km is too rough. First, the first layer is approximately 50m, but it will be usual to set 20-30m. The current model configuration is doubled thickness on first layer, and the representativeness as surface layer is ambiguous. Second, the upper model height is only 10 km. In my best knowledge, CMAQ does not support the top boundary condition. Therefore, this modeling system might have some problem to the treatment of stratospheric O3, and subsequently, to the model performance on surface level. The statistic analysis for O3 (Table S2) seems to be out of range compared to the suggested model performance (Emery et al. 2017). Furthermore, this reproducibility for O3 might lead to inaccuracy of other air pollutants.
   Reference) Emery et al. (2017, JA&WMA)
   https://www.tandfonline.com/doi/full/10.1080/10962247.2016.1265027

5) The emission inventory for Beijing is not taken from MEIC, but there is no reference and needs elevant information here. What was the difference between two inventories? Did the authors have specific reason to replace the emissions only for Beijing instead of MEIC?

6) What was the biomass burning inventory used in this study? I did not find the description.

7) Section 2.4.2: In Table S2, I can only find the statistic for the year of 2017. Why did other years not shown? This section should be clearly separated into the description and discussion. Most of this section should be moved to subsection 3.2 or 3.1.

8) Table 2: Does the parenthesis on rightmost column indicate observation? It should be clearly described.

9) Typo: Section 3.4.2 should be 3.4.3

---

## Referee Comment (RC2) · Anonymous Referee #2 · 21 Jan 2019

This paper systematically quantifies the relative importance of local control measures, surrounding emission reductions and meteorological changes in PM2.5 air quality improvement in Beijing during 2013-2017. A number of sensitivity simulations are performed, which are huge load of work. The paper is generally well written and the conclusions have strong policy implications. I would suggest publishing it after addressing the following issues.

1 The authors provide comprehensive validation of meteorological variables and concentrations of criteria pollutants. It would be nice to include also validation of PM2.5

compositions and draw conclusions on which species are more important for the declines in PM2.5.

2 The description of scenario design and decomposition analysis is very confusing. In equations (2) and (3), i=1..9, but in Table 2, i=1...7. I understand the other two cases are impact of meteorology and emission reduction of surroundings, but it would be better to improve the descriptions here. Additionally, the response of PM2.5 is not linear to emission changes in the inventory, so it might be questionable to sum them up directly in equations (2) and (3).

Minor comments: Page 7 line 11: SIME17S13M17 and SIME17S13M17 typo?

Page 7 line 12: change "In both of these cases" to "in both cases"

---

## Author Comment (AC1) · 4 Mar 2019

Dominant role of emission reduction in PM2.5 air quality improvement in Beijing during 2013-2017: a model-based decomposition analysis by Jing Cheng et al.

General Comments

The main purpose of this paper is the discussion of the reasons of recent rapid PM2.5 decrease of Beijing, mainly by using the meteorological and emission sensitivities by chemical transport model. Based on several sensitivity simulations, this paper made a decomposition analysis framework to evaluate the impacts of local control policies, surrounding emission reductions and the meteorological changes on PM2.5 abatement in Beijing during 2013-2017 and 2016-2017. This paper made the most of important sensitivity analysis and explains the relative contribution of meteorology (12%), local emission (65%) and regional emission (23%) for the reduction of PM2.5 between 2017 and 2013. The results of detailed sensitivity analyses are useful for the understanding of PM2.5 reduction and environmental policy. Their results are very much reasonable and important. However it is difficult to find scientific uniqueness of this paper. The current version of manuscript could be published as the Technical Note. It is necessary to add the more scientific discussion in order to be accepted as a research paper.

**Response:**

**We thank the reviewer #1 for the constructive comments and address them as below.**

Specific Comments

1) The paper mainly discussed the decreases of annual average PM2.5 concentration. However as shown in Figure 4, the authors are specifying the decrease of chemical compositions (i.e., SO2, NOx), it is necessary to compare the aerosol chemical composition changes too. The under estimation of sulfate in winter are reported in many previous papers, the

comparison of sulfate between observation and model results are very much necessary.

**Response: we add the validation of PM$_{2.5}$ compositions simulations in Sect.2.4.2 (the validation is conducted in terms of the absolute concentration values comparison and the temporal change comparison); the detailed comparison of the simulation and observations of PM$_{2.5}$ compositions are listed in SI, Table S4; the data source of observational PM$_{2.5}$ compositions is introduced in Sect.2.1 (SPARTAN observational PM$_{2.5}$ compositions data was the most complete public data we could found, however with certain temporal discontinuities. This may not be able to support the trend analysis, but can used for period validation. Besides this observational data set, we also collected some trends observational analysis in reported research (Shao et al., 2018) to verify the temporal changes of simulated PM$_{2.5}$ compositions).**

**As Table S4 showed, the simulation of sulfate basically agrees with the observation, even with slight overestimation in winter. On the one hand, a missing heterogeneous chemistry mechanism is revised in our model based on our previous work (Zheng et al., 2015), which can improve the sulfate simulation in pollution events. On the other hand, along with the effective SO$_2$ emission control measures in Beijing in recent years, the SO$_4^{2-}$ was basically no longer the key contributor leading to heavy pollution, while the nitrate-driven haze pollution has become more dominate, especially in the summertime (Li et al., 2018). Additionally, the estimation of SO$_2$ emission might be overestimated, for the obvious overestimation of SO$_2$ simulation (SI, Table S3).**

**We also add an analysis about the aerosol chemical composition changes in Sect.3.4.1 as follows, and the variation trends of simulated PM$_{2.5}$ compositions can be found in SI, Figure S6.**

**Added/rewritten part in Sect.2.1:** To evaluate the accuracy of PM$_{2.5}$ components simulation, we collected the PM$_{2.5}$ components data from the Surface PARTiculate mAtter Network (SPARTAN, www.spartan-network.org). SPARTAN is a global, long-term project that observed and analyzed the particulate matter mass, water-soluble ions, black carbon and

[revised manuscript text omitted]

2) Another important point is the OC (we found that model result are usually under estimated), so the detailed examinations of model reproductively of OC are necessary in order to discuss the emission change in VOC.

    **Response: please refer to the response for Specific Comments (1).**

3) The authors show the very good agreement of PM2.5 reduction between observation and model results. It is usually difficult to have such good agreement. It is necessary to discuss the detailed reasons why model results are so good agreement from the view point of emission inventory, WRF model, model horizontal resolution, and CMAQ itself performance.

**Response: In our simulation, a better emission inventory of Beijing (described in Sect.2.2), with more accurate e spatial distribution and emission source allocation, was adopted in our simulation, which can largely improve the air quality modelling, especially when modelling with finer resolutions (Zheng et al., 2017). Meanwhile, analysis nudging, observational nudging and soil nudging were adopted in WRF simulation, which improved the meteorology modelling (SI, Table S2 (a)-(f)). The statistic of WRF simulation results showed a better performance. Additionally, three nested domains were designed and the horizontal resolution of the third domain was 4km × 4km. The higher resolution enhanced the advantage and accuracy of BJ-EI; meanwhile the simulation of finer grids would create higher spatial accuracy and better agree with the observational data. However, the model was not entirely perfect. On the one hand, certain biases existed in the simulated and observational concentrations (such as the underestimation of $PM_{2.5}$, $O_3$ and the overestimation of $SO_2$ (SI, Table S3(b)), especially in some heavy pollution episodes. And the biases became even larger in the $PM_{2.5}$ compositions comparisons (SI, Table S4). On the other hand, the presented validation analysis was the average results from 12 observation stations in Beijing. The simulation (both the reductions and absolute concentrations) of different grids (where different observation stations located) also presented different performance, which can be seen from the certain ME and NME statistics (SI, Table S3, Table S4). Generally speaking, the better emission inventory and higher resolutions in this study led to a better model performance, however certain biases and uncertainties existed due to the missing mechanism in the model itself and some other limitations.**

4) Zero-out emission sensitivity is used in this study by assuming the linearity. PM2.5 formation is usually nonlinear, so it is necessary why the authors are using zero-out method.

**Response: we add a discussion part (Sect.3.5.1) to quantify the extra non-linearity effects of the zero-out approach in our study; meanwhile, we also explained the reason why we used zero-out approach in Sect.3.5.1 as follows.**

**Added/rewritten part in Sect.3.5.1:** Although various methods have been developed to quantify the source of $PM_{2.5}$ and evaluate their contributions, such as receptor-based methods (like CMB and PMF), trajectory-based methods (like PSCF and EEI)), source-oriented methods (like CAMx-PSAT and CMAQ-ISAM)) (Li et al, 2015), they can hardly consider the meteorology and emission changes simultaneously. Therefore, the zero-out approach might be a better choice to attribute the contribution of local and regional emission control as well as meteorology changes under one complete decomposition framework. The zero-out method is also widely used in estimating the contribution of air pollution sources (Lelieveld et al., 2015; Han et al., 2016; Baker et al., 2016; Zhang et al., 2017; Zhang et al., 2017; Ni et al., 2018).

However, the response of $PM_{2.5}$ formulation is not linear to the meteorology and emission changes; thus, the zero-out approach would introduce extra bias in research. The non-linear effects of the analyse period of 2013-2017 could be evaluated by the following equation (Zhang et al., 2017).

$$\text{Bias} = (\text{SCon}(M) + \text{SCon}(S) + \sum_{i=1}^{7} \text{SCon}(pi)) - (\text{SPM}_{2.5}(E_{L13S13}M_{13}) - \text{SPM}_{2.5}(E_{L17S17}M_{17})) \quad (7)$$

Where $SPM_{2.5}(E_{L13S13}M_{13})$ and $SPM_{2.5}(E_{L17S17}M_{17})$ represent the direct simulated $PM_{2.5}$ concentration of base case in 2013 and 2017. The balance of their values is the actual $PM_{2.5}$ decrement during 2013-2017 under the mixed impacts of meteorology change, regional and local emission reductions. The sum of $SCon(M)$, $SCon(S)$ and $\sum_{i=1}^{7} SCon(pi)$ represents a linear result of all contributors during this period. The extra bias can be estimated as the difference between the linear addition and the actual decrement. According to equation (7), we estimated biases in the analyse of 2013-2017 were 1.4 µg m$^{-3}$, accounting for 4.3%. Similarly, the absolute and relative biases in the analysis of 2016-2017 were estimated as -0.6 µg m$^{-3}$ and -3.6%. Both indicated the non-linear effects are relatively small and acceptable.

5) Section 2.3: Although we can follow by Zheng et al. (2018) to understand the emission inventory, it should be noted what is the major "updated". Especially, the inventory is named as "MEIC"; however, we noticed the emission amounts described in Zheng et al. (2018) and Li, M. (2017) are different. Does this mean "updated"? Taking into account the importance of emission inventory, more careful explanations and descriptions are needed here for the traceability of this kind of study.

**Response: The MEIC model is developed by Tsinghua University, and keeps continuous updating. Li, M. (2017) focused on the year of 2008, 2010 and 2012; while Zheng et al. (2018) focused on the periods of 2010-2017, discussing the temporal emission changes in China form 2010-2017. We add some detailed information about Zheng's work to explain the major updated specifics in MEIC model in Sect.2.3 as follows; and a brief introduction of MEIC is also added.**

**Added/rewritten part in Sect.2.3:** The air pollutant emission inventory of Beijing's surrounding regions (including Tianjin, Hebei, Henan, Shandong, Shanxi, Inner Mongolia) for the period of 2013-2017 was obtained from the MEIC model. MEIC is a bottom-up emission inventory model developed for China by Tsinghua University, covering 31 provinces in China and the year range of 1990 up to now. More than 700 emission sources were developed in MEIC model. The methodologies and data on which the MEIC is based, as well as the continues updating process of pollutants emission factors, have been introduced in our previous studies (Zhang et al 2007; Liu et al., 2010; Lei et al., 2011; Li et al., 2014; Shen et al., 2014; Hong et al., 2017; Qi et al., 2017; Zheng et al., 2017; Li et al., 2017a; Li et al., 2017b). For the updated emission inventory and emission reductions of the surrounding regions, we referee to our previous work (Zheng et al., 2018). In their work, Zheng et al collected the latest Chinese Energy Statistics data from National Bureau of Statistics, the industrial production and technology penetration data, and the unpublished data from the Ministry of Ecology and Environment. Then they estimated and updated China's anthropogenic emissions from 2010 to 2017 under the framework of MEIC model. Particularly, they calibrated the emission calculation parameters (such as technology penetrations and removal efficiencies) in major sectors, such as power, cement, steel, iron, of each province based on emission control policies. Based on their updated China

anthropogenic emission inventory, we collected and analyzed the detailed sectoral emission reductions and trends of Beijing's surrounding regions during 2013-2017.

6) Section 2.4.1: Model descriptions are insufficient. Model calculation was conducted after only 10 days spin-up. How does the 3 years WRF simulation perform? i.e., with or without FDDA? Need more detailed descriptions.

**Response: we rewrite and add some detailed descriptions about the WRF-CMAQ model configurations in Sect.2.4.1 (as follows). In our study, we made a complete base simulation of 2013-2017 and obtained a consecutive five-year WRF and CMAQ simulation results. The 10 days spin-up was adopted to 18 sensitivity experiments. Analysis nudging, observational nudging and soil nudging were adopted, and FDDA data was from the U.S. National Centers for Environmental Prediction (NCEP, http://rda.ucar.edu/datasets/), Automated Data Processing surface (ds461.0) and upper (ds351.0) air data. The nudging-related configurations in WRF model were referred to our previous work (Zheng et al., 2015) and presented a good performance in meteorology modelling of North China.**

**Aadded/rewritten part in Sect.2.4.1:** For the WRF model configuration, we chose the New Goddard scheme (Chou et al., 1998) and RRTM (Mlawer et al., 1997) for shortwave and longwave radiation options, the Kain-Fritsch cloud parameterization (version 2, Kain, 2004), the ACM2 PBL scheme (Pleim, 2007), the Pleim-Xiu land-surface scheme (Xiu and Pleim, 2000) and WSM6 cloud microphysics (Hong and Lim, 2006). Analysis nudging, observational nudging and soil nudging were adopted, and FDDA data was from the U.S. National Centers for Environmental Prediction (NCEP, http://rda.ucar.edu/datasets/), Automated Data Processing surface (ds461.0) and upper (ds351.0) air data. The meteorological initial and boundary conditions were derived from the final analysis data (FNL). We made a continuous meteorology simulation during 2013-2017, with a ten-days spin-up before this period.

7) Does CMAQ study include the Asian dust? If yes, need discussion of model accuracy and problems.

**Response: Asian dust was included in this study. We add the dust emission descriptions in Sect.2.4.1 as follows; and we also add a discussion about the relevant simulation results in Sect.2.4.2.**

**Added/rewritten part in Sect.2.4.1:** For the dust emission, bare lands dust was calculated by the in-line windblown dust in CMAQ model. As 2.2 section described, other dust source, such as road dust and construction dust were added in BJ-EI, while missed in MEIC model. It brought uncertainty of air quality simulation, especially for the $PM_{10}$ simulation in other regions.

**Added/rewritten part in Sect.2.4.2:** The added fugitive dust emission (road and construction dust) in BJ-EI has improved the $PM_{10}$ simulation of Beijing, with an overestimation range of 4% - 12% (Table S3 (b)). However, the $PM_{10}$ simulation of other cities in the third domain were underestimated (-8%) - (34%), especially in some heavy industry cities as Tangshan (-34%), Baoding (-25%), and Handan (-29%). This might be attributed to lack of construction dust emissions in these regions, as well as the uncertainty of the dust model (Todd et al., 2008; Foroutan et al., 2018). Similar methods and phenomenon were also applied and reported in previous researches (Wang et al., 2010). Although it might introduce the uncertainty of simulation; given that our research was focused on the attribution analyses of anthropogenic emission changes in Beijing, this uncertainty was relatively small.

8) Emission sensitivity study was conducted by each emission sector base. It is necessary to include the discussion of the accuracy (or error bars) of emission estimate for each sector base.

**Response: we add a discussion part to describe the uncertainty of emission estimations in Sect.3.5.2 (as follows). We first represent the quantified error bars of MEIC model (Zhang et al., 2009) and the uncertainty of regional emission estimations (Zheng et al., 2018) referred to our previous work. Then based on**

**Zheng's method, we discussed the uncertainty of BJ-EI (the local emission inventory of Beijing) (Sect.3.5.2; SI, Figure S5). Finally, we discussed the uncertainty in the estimation of measure-based emission reductions.**

[revised manuscript text omitted]

9) Section 2.5 is unclear. The description of model sensitivity has to be rewrite. Equations of (1) – (3) are unclear. I think Equations of (2) and (3) are not NORMALIZED RESUTS.

**Response: we rewrite the Sect.2.5 (Scenario design and decomposition analysis) as follows:**

**Added/rewritten part in Sect.2.5:** All scenario cases were labelled as $E_{LiSj}M_k$ . $M_k$ (k) represents the metalogical period the case adopted and $E_{LiSj}(i, j)$ represents the emission period. Total emission inventories of China consisted of two parts, that the BJ-EI from BMEMC and the regional (all parts of China except for Beijing) emission inventories from MEIC model. The adopted emission period of these two parts were labelled as $Li(i)$ and $Sj(j)$ respectively. $E_{L13S13}M_{13}$, $E_{L16S16}M_{16}$, and $E_{L17S17}M_{17}$ were three base cases and driven by the actual emission inventories and meteorology of 2013, 2016 and 2017, respectively, to reproduce the air quality of the corresponding year. $E_{L17S17}M_{13}$ and $E_{L17S17}M_{16}$ were designed to investigate the impact of meteorology. These two cases were driven by varying meteorological conditions (meteorology of 2013 and 2016, respectively) and the same emission inventory (for the year 2017). $E_{L17S13}M_{17}$ and $E_{L17S16}M_{17}$ were designed to quantify the impact of surrounding emission reduction during 2013-2017 and 2016-2017. In these two cases, the emission inventory of Beijing was set to the 2017 level, while the regional emission inventory was set to the 2013 and 2016 levels, respectively.

Another fourteen simulations were designed to quantify the air quality improvements contributed by seven types of local control policies during two periods. Cases for 2013-2017 and 2016-2017 were labelled as $E_{L_{pi}S17}M_{17}$ and $E_{L_{qi}S17}M_{17}$ respectively, where $i$ represents the number of each policy (described and listed in Table 1). The meteorological conditions and regional emission inventories of these fourteen cases were set to 2017. For each simulation, emission reduction introduced by the corresponding policy type and adopting period was added to the 2017 baseline, equivalent of "turning off" this type of policy during this period. And then the derived emission inventory was applied to drive the corresponding air quality modelling.

A linear additive relationship was assumed among all contributors to perform a decomposition analysis, and the simulated contributions of all sensitivity cases were then normalized by the difference in observed PM$_{2.5}$ concentrations from 2013-2017 and 2016-2017. The normalization process of 2013-2017 period were calculated by the following equations, while the simulated results for period of 2016-2017 can be normalized with the similar process.

$$\mathrm{SCon(M)} = \mathrm{SPM_{2.5}}(E_{L17S17}M_{13}) - \mathrm{SPM_{2.5}}(E_{L17S17}M_{17}) \tag{1}$$

$$\mathrm{SCon(S)} = \mathrm{SPM_{2.5}}(E_{L17S13}M_{17}) - \mathrm{SPM_{2.5}}(E_{L17S17}M_{17}) \tag{2}$$

$$\mathrm{SCon(pi)} = \mathrm{SPM_{2.5}}(E_{L_{pi}S17}M_{17}) - \mathrm{SPM_{2.5}}(E_{L17S17}M_{17}) \tag{3}$$

$$\mathrm{NCon(M)} = (\mathrm{PM_{2.5OBS2013}} - \mathrm{PM_{2.5OBS2017}}) \times \frac{\mathrm{SCon(M)}}{\mathrm{SCon(M)} + \mathrm{SCon(S)} + \sum_{i=1}^{7}\mathrm{SCon(pi)}} \tag{4}$$

$$\mathrm{NCon(S)} = (\mathrm{PM_{2.5OBS2013}} - \mathrm{PM_{2.5OBS2017}}) \times \frac{\mathrm{SCon(S)}}{\mathrm{SCon(M)} + \mathrm{SCon(S)} + \sum_{i=1}^{7}\mathrm{SCon(pi)}} \tag{5}$$

$$\mathrm{NCon(pi)} = (\mathrm{PM_{2.5OBS2013}} - \mathrm{PM_{2.5OBS2017}}) \times \frac{\mathrm{SCon(pi)}}{\mathrm{SCon(M)} + \mathrm{SCon(S)} + \sum_{i=1}^{7}\mathrm{SCon(pi)}} \tag{6}$$

where $SCon(M)$ represents the simulated contribution of meteorology change during 2013-2017, which equals the balance of simulated PM$_{2.5}$ ($\mu$g m$^{-3}$) from case $E_{L17S17}M_{13}$ and case $E_{L17S17}M_{17}$. Similarly, $SCon(M)$ and $SCon(pi)$ represent the simulated contribution of regional emission reductions and each local control policy type. $NCon(M)$ represents the normalized contribution of meteorology change during 2013-2017, which equals the product of the observational PM$_{2.5}$ balance (from 2013-2017) and the proportion of simulated meteorology contribution (in the simulated contributions of all factors). Similarly, $NCon(M)$ and $NCon(pi)$ represent the normalized contribution of regional emission reductions and each local control policy type.

10) Section 3.2 Without the model evaluation from 2013 to 2017, it is hard to discuss the source attribution results by model. From the current

manuscript, we can only find time-series on 2013, 2016, and 2017 for PM2.5 and statistic evaluation only on 2017. The model evaluation is inadequate at the current manuscript. For example, Figure 3 and Figure S3 can be presented in the same form for model. See also my minor comments 4) for O3 performance.

**Response: we add the consecutive evaluation (2013-2017) of WRF and CMAQ simulation results in SI (Table S2 (a)-(f) for WRF validation; Table S3 (a) for monthly descriptive statistic of $PM_{2.5}$ simulation; Table S4 for validation of $PM_{2.5}$ compositions; and Table S3 (b) for annually statistic of six major pollutants). Relative analysis is also added in Sect.2.4.2 (as follows).**

**Added/rewritten part in Sect.2.4.2:** The time series and evaluation results indicated that the CMAQ model and simulation results in this work can relatively well reproduce the temporal and spatial distribution of air pollutants in Beijing and its surroundings. As for the simulated $PM_{2.5}$ of 2017, the monthly Corr of $PM_{2.5}$ concentrations varied from 0.53 (in May) to 0.89 (in October), and the annual Corr of $PM_{2.5}$ concentrations varied from 0.65 (in 2016) to 0.81 (in 2014). The NMB and NME of monthly $PM_{2.5}$ simulations were within $\pm 45\%$ and $\pm 55\%$, respectively. According to the observation data, the annual average $PM_{2.5}$ concentrations in Beijing decreased by 31.5 $\mu g\ m^{-3}$ from 2013 to 2017, while the simulated $PM_{2.5}$ decreased by 32.8 $\mu g\ m^{-3}$ (Table 2). Compared with 2016, the observed and simulated $PM_{2.5}$ decreased by 14.9 and 16.6 $\mu g\ m^{-3}$, respectively (Table 2). The evaluation results suggested that the modelling system in this work can be used to quantify and analyses the attribution of $PM_{2.5}$ mitigations in Beijing. As for the simulation results of other pollutants in Beijing, the Corr varied from 0.61-0.74 for $SO_2$, 0.59-0.68 for $NO_2$, 0.62-0.78 for CO, 0.64-0.74 for $O_3$, 0.62-0.74 for $PM_{10}$ (Table S3 (b)), which was acceptable for the research. The $SO_2$ simulation was overestimated in the five years, especially during 2013-2015, which indicated the $SO_2$ emission in BJ-EI might be higher than the reality. The added fugitive dust emission (road and construction dust) in BJ-EI has improved the $PM_{10}$ simulation of Beijing obviously, with an overestimation range of 4% - 12% (Table S3 (b)). However, the $PM_{10}$ simulation of other cities in the third domain were underestimated (-8%) - (34%), especially in some heavy industry cities as Tangshan (-34%), Baoding (-25%), and Handan (-29%). This might be attributed to lack of construction and road dust emissions in these regions, as well as the uncertainty of the dust model (Todd et al., 2008; Foroutan et al., 2018). It might introduce

the uncertainty of simulation, but given that our research was focused on the attribution analyses of anthropogenic emission changes in Beijing, this uncertainty was relatively small. The $O_3$ was underestimated in this WRF-CMAQ model system, with the range of (-8.3%) - (-22.6%). The rough vertical layers, the underestimation of nature source emissions, the defect of upper boundary simulation in regional model and the uncertainties of VOCs emission inventories might all lead this underestimation.

11) The results shown here should be interpreted in depth. On 2013, especially the peak on January, model sometimes overestimated observed PM2.5. However, model simulated same level or sometimes underestimated high concentrations during winter on 2016 and 2017. Actually, the model negative-bias is larger in 2017 compared to 2013 (Table 2). Therefore, the source attribution results based on scenario analysis adopted in this study can be strongly reflected by emission variation rather than the observed facts.

Response: more detailed validation analysis is added in Sect.2.4.2, SI, Table S2-Table S4 (please refer to the response for Specific Comments (1), (3), (10), and Minor Comments (4)). The instability of high concentrations simulation might largely attribute to the variation of emission during 2013-2017. In 2013, all pollutants basically emitted at a high level, while decreased remarkably in 2017. The emission change and the relevant response in CMAQ model, as well as the impacts of meteorology, might lead to this instability.

As this comment pointed out, the source attribution results based on scenario analysis adopted in this study can be strongly reflected by emission variation. Therefore, we add a discussion part in Sect.3.5.3 to analysis the uncertainty of emission estimation.

12) Figures 5 and 9 are unfriendly. It need more detailed explanations or improve the presentation of figures. The basic information is same as

Figure 10, so it might be better to modify Figures 5 and 9 into Figure 10 format.

**Response: we updated the Figure 5 and 9 based on Figure 10 format, and add some extra explanation.**

13) Section 3.3: The discussion in this section needs relevant references.

**Response: We add some relevant references in this part. Additionally, the discussion of this section is developed based on Zheng's work (Zheng et al., 2018). In their work, Zheng et al updated China's emission during 2010-2017 based on latest national statistics and relevant control policies. The mainly updates are briefly introduced in Sect. 2.3, and the relevant uncertainty is added in Sect.3.5.2. Based on this China emission inventory, we extract the emissions of Beijing's regional areas, including Tianjin, Hebei, Henan, Shandong, Shanxi and Inner Mongolia, and then analyze the emission variation of these areas during 2013-2017. The emission results are consistent with Zheng's work, while the relevant policy and attribution analysis is based on national clean air action plan.**

Minor Comments

1) It have been reported that WRF should be updated version 3.9.0.1 or later for the upgraded NCEP dataset after 12 UTC, 19 July 2017. http://www2.mmm.ucar.edu/wrf/users/wpsv3.9/known-prob-3.9.html The used version is 3.8, but how did the authors solve this problem?

**Response: we used the WRF version of 3.8 and the WPS version of 3.9.1. Given that the forward compatibility, the results of WPS 3.9.1 can be used to drive WRF3.8 simulations.**

2) What is the horizontal grid resolution in second domain?

**Response: the horizontal grid resolution in second domain is 12km × 12km, we also add this description in Sect.2.4.1.**

3) What is the lateral boundary condition for first domain? It will be taken from global chemical transport model, but did the global model consider year-to-year emission variation? If not, how can we conclude the importance of global-scale impacts on the air quality in China?

**Response: the chemical initial and boundary conditions (ICBCs) of the first domain were interpolated from the output of GEOS-Chem model (Bay et al., 2001; Geng et al., 2015). The year-to-year global emission variation was provided by the emission inventories and scale factors from GEOS-Chem Shared Data Directories (http://acmg.seas.harvard.edu/geos/doc/man/). However, given that our research is focused more on Beijing, the nation-scale impacts, especially the impacts of emission reductions of the surrounding provinces, are much larger than global-scale impacts.**

4) Considering the current modeling application over East Asia, the vertical 14 layers from surface to 10 km is too rough. First, the first layer is approximately 50m, but it will be usual to set 20-30m. The current model configuration is doubled thickness on first layer, and the representativeness as surface layer is ambiguous. Second, the upper model height is only 10 km. In my best knowledge, CMAQ does not support the top boundary condition. Therefore, this modeling system might have some problem to the treatment of stratospheric O3, and subsequently, to the model performance on surface level. The statistic analysis for O3 (Table S2) seems to be out of range compared to the suggested model performance (Emery et al. 2017). Furthermore, this reproducibility for O3 might lead to inaccuracy of other air pollutants. Reference) Emery et al. (2017, JA&WMA)
https://www.tandfonline.com/doi/full/10.1080/10962247.2016.1265027

**Response: In previous manuscript, 14 layers refer to the vertical resolution of CMAQ model, while for WRF was 23 layers. We rewrite this part in Sect.2.4.1 and add some detailed information as follows. Similar vertical configurations for WRF and CMAQ model were applied in a lot of reported studies, and exhibited good agreement of PM$_{2.5}$ simulation with observations (Wang et al., 2010; Xing et al., 2011; Wang et al., 2012; Zhao et al., 2013; Zheng et al., 2015; Cai et al., 2017; Zheng et al., 2017; Campbell et al., 2018).**

**Added/rewritten part in Sect.2.4.1:** The vertical resolution was designed as 23 sigma levels from surface to tropopause (about 100mb) for WRF simulation (with 10 layers below 3-km), while collapsed into 14 CTM layers by Meteorology-Chemistry Interface Processor (MCIP). The 14 sigma levels for CMAQ model vertical resolution were 1.000, 0.995, 0.988, 0.980, 0.970, 0.956, 0.938, 0.893, 0.839, 0.777, 0.702, 0.582, 0.400, 0.200 and 0.000.

**As this comment pointed out, O$_3$ was underestimated in this WRF-CMAQ model system, with the range of (-8.3%) - (-22.6%). The rough vertical layers, the underestimation of nature source emissions, the defect of upper boundary simulation in regional model and the uncertainties of VOCs emission inventories might all lead this underestimation. Given that the PM$_{2.5}$ concentration is the most important target in APPCAP, as well as the major pollutants this research focused on, the uncertainty introduced by O$_3$ simulation is relatively small, but further studies and efforts should made to improve the O$_3$ simulation. We also add a discussion in Sect.3.5.3 for this uncertainty.**

5) The emission inventory for Beijing is not taken from MEIC, but there is no reference and needs elevant information here. What was the difference between two inventories? Did the authors have specific reason to replace the emissions only for Beijing instead of MEIC?

**Response: the emission inventory for Beijing (BJ-EI) in this study is from the Beijing Municipal Environmental Monitoring Center (BMEMC). We add a detailed description of BJ-EI in Sect.2.2; compare the major difference on**

**activity rate and spatial distribution of BJ-EI and MEIC in SI, Table S1; and explained why we replace the BJ-EI of MEIC in Sect.2.2 as follows. Mainly because the spatial distribution and emission source allocation of BJ-EI were more accurate than those of the MEIC model, which can significantly improve the air quality modelling, especially when modelling with finer resolutions (Zheng et al., 2017). Meanwhile, more detailed and objective activity rate, technology distribution and removal efficacy data at the county-level were collected from BJ-EI, which can largely reduce the uncertainty in estimating the emission reductions of each local control policy. However, similar local emission inventory as BJ-EI in other regions are not available, thus only BJ-EI can be replaced in MEIC.**

**Added/rewritten part in Sect.2.2:** In this study, the anthropogenic emission inventory of Beijing was provided by the Beijing Municipal Environmental Monitoring Center (BMEMC). Based on the bottom-up method, the BMEMC developed a high-resolution emission inventory for Beijing (BJ-EI) of 2013 and 2017. BJ-EI basically had the same source classification as the MEIC model (described in Sect.2.3), however, the investigation and calculation process of BJ-EI were conducted at the county level, which of MEIC were conducted at the provincial level. The power, heating, industry (such as cement, iron, steel, chemical industry, manufacturing industry) and most solvent use (such as vehicle paint, ink, paint and coating) sectors were treated as point sources, with a higher accuracy of emission facility locations. In addition, fugitive dust emissions, including bare soil dust, road dust and construction dust, were added in BJ-EI, which were missing in MEIC model because of the lack of activity rates data. More detailed comparisons of BJ-EI and MEIC model can be found in SI, Table S1. Therefore, the spatial distribution and emission source allocation of BJ-EI were more accurate than those of the MEIC model, which can significantly improve the air quality modelling, especially when modelling with finer resolutions (Zheng et al., 2017). Meanwhile, more detailed and objective activity rate, technology distribution and removal efficacy data at the county-level were collected from BJ-EI, which can largely reduce the uncertainty in estimating the emission reductions of each local control policy.

6) What was the biomass burning inventory used in this study? I did not find the description.

**Response: The open biomass burning inventory was not included in this study. On the one hand, the yearly emission changes are important in this study, however, current open biomass burning products (such as the Global Fire Emissions Database (van et al., 2017; http://www.globalfiredata.org/), the emission inventory of crop burning in China (Huang et al., 2012)) could not provide year-to-year emission variations, and the open biomass burning emission inventory for recent years was not available. On the other hand, although the lack of open biomass burning might introduce some uncertainties for the research, given that we mainly focused on the attribution of anthropogenic emission changes and meteorology impacts, the uncertainty introduced by this lack might be relatively small. We also add a discussion in Sect.3.5.3 for this uncertainty.**

7) Section 2.4.2: In Table S2, I can only find the statistic for the year of 2017. Why did other years not shown? This section should be clearly separated into the description and discussion. Most of this section should be moved to subsection 3.2 or 3.1.

**Response: please refer to the response for Specific Comments (10). We add the consecutive monthly descriptive statistic of $PM_{2.5}$ simulation during 2013-2017 in TableS3 (a), $PM_{2.5}$ compositions validation in Table S4 and other pollutants validations in Table S3 (b). We also add the analysis of these descriptive statistics in Sect.2.4.2.**

8) Table 2: Does the parenthesis on rightmost column indicate observation? It should be clearly described.

**Response: thanks for the kind remind. The parenthesis on rightmost column indicate the annual mean observational $PM_{2.5}$ concentrations of 2013, 2016 and 2017.**

9) Typo: Section 3.4.2 should be 3.4.3

**Thanks for the kind remind; and the error is corrected in the new version.**

**References: (the references in bold texts are listed as follows, while the references in the added/written parts can be found in the manuscript)**

Bey, I., D. J. Jacob., R. M. Yantosca., J. A. Logan., B. D. Field., A. M. Fiore., Q. Li., H. Y. Liu., L. J. Mickley., and M. G. Schultz.: Global modeling of tropospheric chemistry with assimilated meteorology: Model description and evaluation, J. Geophys. Res., 106, D19, 23073–23095, https://doi.org/10.1029/2001JD000807, 2001.

Campbell, P., Zhang Y., Yan, F., Lu, Z., and Streets, D.: Impacts of transportation sector emissions on future U.S. air quality in a changing climate. Part I: Projected emissions, simulation design, and model evaluation, Environmental Pollution., 238, 903-917, https://doi.org/10.1016/j.envpol.2018.04.020, 2018.

Geng, G., Zhang, Q., Martin, R.V., Donkelaar, A. v., Huo, H., Che, H., Lin, J., Xin, J., He, K.: Estimating ground-level $PM_{2.5}$ concentration in China from satellite-based aerosol optical depth and chemical transport model. Remote. Sens. Environ., 166, 262-270, https://doi.org/10.1016/j.rse.2015.05.016, 2015.

Huang, X., Li, M., Li, J., and Song, Y.: A high-resolution emission inventory of crop burning in fields in China based on MODIS thermal anomalies/fire products, Atmos. Environ., 50, 9-15, https://doi.org/10.1016/j.atmosenv.2012.01.017, 2012.

Li, H., Zhang, Q., Zheng, B., Chen, C., Wu, N., Guo, H., Zhang, Y., Zheng, Y., Li, X., and He, K.: Nitrate-driven urban haze pollution during summertime over the North China Plain, Atmos. Chem. Phys., 18, 5293-5306, https://doi.org/10.5194/acp-18-5293-2018, 2018.

Li, M., Zhang, Q., Kurokawa, J.-I., Woo, J.-H., He, K., Lu, Z., Ohara, T., Song, Y., Streets, D. G., Carmichael, G. R., Cheng, Y., Hong, C., Huo, H., Jiang, X., Kang, S., Liu, F., Su, H., and Zheng, B.: MIX: a mosaic Asian anthropogenic emission inventory under the international collaboration framework of the MICS-Asia and HTAP, Atmos. Chem. Phys., 17, 935-963, https://doi.org/10.5194/acp-17-935-2017, 2017a.

Shao, P., Tian, H., Sun, Y., Liu, H., Wu, B., Liu, S., Liu, X., Wu,Y., Liang, W., Wang, Y., Gao, J., Xue, Y., Bai, X., Liu, W., Lin,S., and Hu, G.: Characterizing remarkable changes of severe haze events and chemical compositions in multi-size airborne particles (PM1, PM2:5 and PM10) from January 2013 to 2016–2017 winter in Beijing, China, Atmos. Environ., 189, 133–144, https://doi.org/10.1016/j.atmosenv.2018.06.038, 2018.

van der Werf, G. R., Randerson, J. T., Giglio, L., Collatz, G. J., Mu, M., Kasibhatla, P. S., Morton, D. C., DeFries, R. S., Jin, Y., and van Leeuwen, T. T.: Global fire emissions and the contribution of deforestation, savanna, forest, agricultural, and peat fires (1997–2009), Atmos. Chem. Phys., 10, 11707–11735, https://doi.org/10.5194/acp-10-11707-2010, 2010.

Wang, L., Xu, J., Yang, J., Zhao, X., Wei, W., Cheng, D., Pan, X., and Su, J.: Understanding haze pollution over the southern Hebei area of China using the CMAQ model, Atmos. Environ., 56, 69-79, https://doi.org/10.1016/j.atmosenv.2012.04.013, 2012.

Wang, S., Zhao, M., Xing, J., Wu, Y., Zhou, Y., Lei, Y., He, K., Fu, L., and Hao, J.: Quantifying the Air Pollutants Emission Reduction during the 2008 Olympic Games in Beijing, Environ. Sci. Technol., 44, 7,2490-2496, https://doi.org/10.1021/es9028167, 2010.

Xing, J., Zhang, Y., Wang, S., Liu, X., Cheng, S., Zhang, Q., Chen, Y., Streets, D., Jang, C., Hao, J., and Wang, W.: Modeling study on the air quality impacts from emission reductions and atypical meteorological conditions during the 2008 Beijing Olympics, Atmos. Environ., 45, 10, 1786-1798, https://doi.org/10.1016/j.atmosenv.2011.01.025, 2011.

Zhang, Q., Streets, D. G., Carmichael, G. R., He, K. B., Huo, H., Kannari, A., Klimont, Z., Park, I. S., Reddy, S., Fu, J. S., Chen, D., Duan, L., Lei, Y., Wang, L. T., and Yao, Z. L.: Asian emissions in 2006 for the NASA INTEX-B mission, Atmos. Chem. Phys., 9, 5131-5153, https://doi.org/10.5194/acp-9-5131-2009, 2009.

Zhao, B., Wang, S., Wang, J., Fu, J. S., Liu, T., Xu, J., Fu, X., and Hao, J.: Impact of national $NO_x$ and $SO_2$ control policies on particulate matter pollution in China, Atmos. Environ., 77, 453-463, https://doi.org/10.1016/j.atmosenv.2013.05.012, 2013.

Zhao, H., Zheng, Y., He, K., and Zhang, Q.: Trends in China's anthropogenic emissions since 2010 as the consequence of clean air actions, Atmos. Chem. Phys., 18, 14095-14111, https://doi.org/10.5194/acp-18-14095-2018, 2018.

Zheng, B., Zhang, Q., Tong, D., Chen, C., Hong, C., Li, M., Geng, G., Lei, Y., Huo, H., and He, K.: Resolution dependence of uncertainties in gridded emission inventories: a case study in Hebei, China, Atmos. Chem. Phys., 17, 921-933, https://doi.org/10.5194/acp-17-921-2017, 2017.

Zheng, Y., Xue, T., Zhang, Q., Zhang, Geng, G., Tong, D., Li, X., and He, K.: Air quality improvements and health benefits from China's clean air action since 2013, Environ. Res. Lett., 12, 11, 114020, https://doi.org/10.1088%2F1748-9326%2Faa8a32, 2017.

Zheng, B., Tong, D., Li, M., Liu, F., Hong, C., Geng, G., Li, H., Li, X., Peng, L., Qi, J., Yan, L., Zhang, Y., Zhao, H., Zheng, Y., He, K., and Zhang, Q.: Trends in China's anthropogenic emissions since 2010 as the consequence of clean air actions, Atmos. Chem. Phys., 18, 14095-14111, https://doi.org/10.5194/acp-18-14095-2018, 2018.

Zheng, B., Zhang, Q., Zhang, Y., He, K. B., Wang, K., Zheng, G. J., Duan, F. K., Ma, Y. L., and Kimoto, T.: Heterogeneous chemistry: a mechanism missing in current models to explain secondary inorganic aerosol formation during the January 2013 haze episode in North China, Atmos. Chem. Phys., 15, 2031-2049, https://doi.org/10.5194/acp-15-2031-2015, 2015.

---

## Author Comment (AC2) · 4 Mar 2019

This paper systematically quantifies the relative importance of local control measures, surrounding emission reductions and meteorological changes in PM2.5 air quality improvement in Beijing during 2013-2017. A number of sensitivity simulations are performed, which are huge load of work. The paper is generally well written and the conclusions have strong policy implications. I would suggest publishing it after addressing the following issues.

**Response:**

**We thank the reviewer #2 for the constructive comments and address them as below.**

1 The authors provide comprehensive validation of meteorological variables and concentrations of criteria pollutants. It would be nice to include also validation of $PM_{2.5}$ compositions and draw conclusions on which species are more important for the declines in $PM_{2.5}$.

**Response: to better analysis the validation of $PM_{2.5}$ compositions, we firstly add the validation of $PM_{2.5}$ compositions simulations in Sect.2.4.2; the detailed comparison of the simulation and observations of $PM_{2.5}$ compositions are listed in SI, Table S4; the data source of observational $PM_{2.5}$ compositions is introduced in Sect.2.1. Then the analysis about the aerosol chemical composition changes is added in Sect.3.4.1 as follows, and the variation trends of simulated $PM_{2.5}$ compositions can be found in SI, Figure S6.**

Although there was a steady decline in PM$_{2.5}$ concentrations of Beijing during 2013-2017, the trends of PM$_{2.5}$ compositions varied differently. The simulation results of base cases (which adopted the real meteorology and emissions of each year) showed that the sulfate (SO$_4^{2-}$) and organic matter (OM) were dominate species for the decline in PM$_{2.5}$ concentrations during 2013-2017, with the decrement of 7.5 μg m$^{-3}$ (56.6%) and 9.6 μg m$^{-3}$ (40.5%) respectively. The contribution of SO$_4^{2-}$ to the total PM$_{2.5}$ also decreased obviously, from 15.3% in 2013 to 10.7% in 2017; and OM proportion decreased from 27.5% in 2013 to 26.5% in 2017. The rapid decrement of SO$_4^{2-}$ was consistent with the remarkable SO$_2$ emission reductions in Beijing during 2013-2017. Along with the effective SO$_2$ emission control measures, the SO$_4^{2-}$ was basically no longer the key contributor leading to heavy pollution in Beijing while the nitrate-driven haze pollution has become more dominate in Beijing in recent years, especially in the summertime (Li et al., 2018). The decrement of OM was mainly caused by the prominent emission reductions of primary organic carbon (mainly come from the residential burning and other coal combustion sources). VOCs emission reductions also contributed to the OM decreasing, however, due to the insufficient simulation of secondary organic aerosols (SOA) formulations in CMAQ model, the contributions of VOCs emission control might be underestimated. In contrast, nitrate (NO$_3^-$) increased in 2014-2016, and kept basically the same concentration level in 2017 (10.4 μg m$^{-3}$) as 2013 (10.9 μg m$^{-3}$). However, the contribution of NO$_3^-$ to the total PM$_{2.5}$ increased a lot, from 12.7% in 2013 to 19.4% in 2017. The specific concentration and proportion trends of PM$_{2.5}$ concentrations can be found in SI, Table S6.

2 The description of scenario design and decomposition analysis is very confusing. In equations (2) and (3), i=1...9, but in Table 2, i=1…7. I understand the other two cases are impact of meteorology and emission reduction of surroundings, but it would be better to improve the descriptions here. Additionally, the response of PM$_{2.5}$ is not linear to emission changes in the inventory, so it might be questionable to sum them up directly in equations (2) and (3).

**Response: 1) we rewrite the Sect.2.5 (Scenario design and decomposition analysis) as follows:**

All scenario cases were labelled as $E_{LiSj}M_k$. $M_k$(k) represents the metalogical period the case adopted

and $E_{LiSj}(i, j)$ represents the emission period. Total emission inventories of China consisted of two parts, that the BJ-EI from BMEMC and the regional (all parts of China except for Beijing) emission inventories from MEIC model. The adopted emission period of these two parts were labelled as $Li(i)$ and $Sj(j)$ respectively.

$E_{L13S13}M_{13}$, $E_{L16S16}M_{16}$, and $E_{L17S17}M_{17}$ were three base cases and driven by the actual emission inventories and meteorology of 2013, 2016 and 2017, respectively, to reproduce the air quality of the corresponding year. $E_{L17S17}M_{13}$ and $E_{L17S17}M_{16}$ were designed to investigate the impact of meteorology. These two cases were driven by varying meteorological conditions (meteorology of 2013 and 2016, respectively) and the same emission inventory (for the year 2017). $E_{L17S13}M_{17}$ and $E_{L17S16}M_{17}$ were designed to quantify the impact of surrounding emission reduction during 2013-2017 and 2016-2017. In these two cases, the emission inventory of Beijing was set to the 2017 level, while the regional emission inventory was set to the 2013 and 2016 levels, respectively.

Another fourteen simulations were designed to quantify the air quality improvements contributed by seven types of local control policies during two periods. Cases for 2013-2017 and 2016-2017 were labelled as $E_{L_{pi}S17}M_{17}$ and $E_{L_{qi}S17}M_{17}$ respectively, where $i$ represents the number of each policy (described and listed in Table 1). The meteorological conditions and regional emission inventories of these fourteen cases were set to 2017. For each simulation, emission reduction introduced by the corresponding policy type and adopting period was added to the 2017 baseline, equivalent of "turning off" this type of policy during this period. And then the derived emission inventory was applied to drive the corresponding air quality modelling.

A linear additive relationship was assumed among all contributors to perform a decomposition analysis, and the simulated contributions of all sensitivity cases were then normalized by the difference in observed PM$_{2.5}$ concentrations from 2013-2017 and 2016-2017. The normalization process of 2013-2017 period were calculated by the following equations, while the simulated results for period of 2016-2017 can be normalized with the similar process.

$$SCon(M) = SPM_{2.5}(E_{L17S17}M_{13}) - SPM_{2.5}(E_{L17S17}M_{17}) \qquad (1)$$

$$SCon(S) = SPM_{2.5}(E_{L17S13}M_{17}) - SPM_{2.5}(E_{L17S17}M_{17}) \qquad (2)$$

$$SCon(pi) = SPM_{2.5}(E_{L_{pi}S17}M_{17}) - SPM_{2.5}(E_{L17S17}M_{17}) \tag{3}$$

$$NCon(M) = (PM_{2.5OBS2013} - PM_{2.5OBS2017}) \times \frac{SCon(M)}{SCon(M) + SCon(S) + \sum_{i=1}^{7} SCon(pi)} \tag{4}$$

$$NCon(S) = (PM_{2.5OBS2013} - PM_{2.5OBS2017}) \times \frac{SCon(S)}{SCon(M) + SCon(S) + \sum_{i=1}^{7} SCon(pi)} \tag{5}$$

$$NCon(pi) = (PM_{2.5OBS2013} - PM_{2.5OBS2017}) \times \frac{SCon(pi)}{SCon(M) + SCon(S) + \sum_{i=1}^{7} SCon(pi)} \tag{6}$$

where $SCon(M)$ represents the simulated contribution of meteorology change during 2013-2017, which equals the balance of simulated PM$_{2.5}$ (μg m$^{-3}$) from case $E_{L17S17}M_{13}$ and case $E_{L17S17}M_{17}$. Similarly, $SCon(M)$ and $SCon(pi)$ represent the simulated contribution of regional emission reductions and each local control policy type. $NCon(M)$ represents the normalized contribution of meteorology change during 2013-2017, which equals the product of the observational PM$_{2.5}$ balance (from 2013-2017) and the proportion of simulated meteorology contribution (in the simulated contributions of all factors). Similarly, $NCon(M)$ and $NCon(pi)$ represent the normalized contribution of regional emission reductions and each local control policy type.

**2) we add a discussion part in Sect.3.5.1 to quantify the extra non-linearity effects of the zero-out approach in our study; meanwhile, we also explained the reason why we used zero-out approach in Sect.3.5.1 (as follows).**

Although various methods have been developed to quantify the source of PM$_{2.5}$ and evaluate their contributions, such as receptor-based methods (like CMB and PMF), trajectory-based methods (like PSCF and EEI)), source-oriented methods (like CAMx-PSAT and CMAQ-ISAM)) (Li et al, 2015), they can hardly consider the meteorology and emission changes simultaneously. Therefore, the zero-out approach might be a better choice to attribute the contribution of local and regional emission control as well as meteorology changes under one complete decomposition framework. The zero-out method is also widely used in estimating the contribution of air pollution sources (Lelieveld et al., 2015; Han et al., 2016; Baker et al., 2016; Zhang et al., 2017; Zhang et al., 2017; Ni et al., 2018).

However, the response of PM$_{2.5}$ formulation is not linear to the meteorology and emission changes; thus, the zero-out approach would introduce extra bias in research. The non-linear effects of the analyse period of 2013-2017 could be evaluated by the following equation (Zhang et al., 2017).

$$\text{Bias} = (\text{SCon(M)} + \text{SCon(S)} + \sum_{i=1}^{7} \text{SCon(pi)}) - (\text{SPM}_{2.5}(E_{L13S13}M_{13}) - \text{SPM}_{2.5}(E_{L17S17}M_{17})) \qquad (7)$$

Where $SPM_{2.5}(E_{L13S13}M_{13})$ and $SPM_{2.5}(E_{L17S17}M_{17})$ represent the direct simulated PM$_{2.5}$ concentration of base case in 2013 and 2017. The balance of their values is the actual PM$_{2.5}$ decrement during 2013-2017 under the mixed impacts of meteorology change, regional and local emission reductions. The sum of $SCon(M)$, $SCon(S)$ and $\sum_{i=1}^{7} SCon(pi)$ represents a linear result of all contributors during this period. The extra bias can be estimated as the difference between the linear addition and the actual decrement. According to equation (7), we estimated biases in the analyse of 2013-2017 were 1.4 µg m$^{-3}$, accounting for 4.3%. Similarly, the absolute and relative biases in the analysis of 2016-2017 were estimated as -0.6 µg m$^{-3}$ and -3.6%. Both indicated the non-linear effects are relatively small and acceptable.

Minor comments: Page 7 line 11: SIME17S13M17 and SIME17S13M17 typo?

**Response:** SIME17S13M17 **represents the simulation that adopted the meteorology of 2017, Beijing local emission of 2017, Beijing surrounding emission of 2013. In the previous version manuscript, "Page 7 line 11:** SIME17S13M17 **and** SIME17S13M17**", the second** SIME17S13M17 **was wrong and should be** SIME17S13M17, **which represents the simulation that adopted the meteorology of 2017, Beijing local emission of 2017, Beijing surrounding emission of 2016. This section is rewritten now, and please refer to the response of comment 1.**

Page 7 line 12: change "In both of these cases" to "in both cases"

**Thanks for the kind remind; and the error is corrected in the new version.**